# The Rab7-Epg5 and Rab39-ema modules cooperatively position autophagosomes for efficient lysosomal fusions

Attila Boda[1,2], Villő Balázs[1,2], Anikó Nagy[1,2], Dávid Hargitai[1,2], Mónika Lippai[1,2], Zsófia Simon-Vecsei[1,2], Márton Molnár[1,2], Fanni Fürstenhoffer[1,2], Gábor Juhász[1,3], Péter Lőrincz[1,2]*

[1]Department of Anatomy, Cell and Developmental Biology, Eötvös Loránd University, Budapest, Hungary; [2]HAS-ELTE Momentum Vesicular Transport Research Group, Hungarian Academy of Sciences & ELTE Eötvös Loránd University, Budapest, Hungary; [3]Lysosomal Degradation Research Group, Institute of Genetics, HUN-REN BRC Szeged, Szeged, Hungary

*For correspondence:
peter.lorincz@ttk.elte.hu

Competing interest: The authors declare that no competing interests exist.

## eLife Assessment

This paper presents **valuable** findings on how autophagosomes are positioned along microtubules for their efficient fusion with lysosomes, providing significant insights into the mechanism. The evidence supporting the conclusions is **solid**, with high-quality fluorescence microscopy combined with Drosophila genetics. This work will be of broad interest to cell biologists interested in autophagy and related cell biology fields.

**Abstract** Macroautophagy, a major self-degradation pathway in eukaryotic cells, utilizes autophagosomes to transport self-material to lysosomes for degradation. While microtubular transport is crucial for the proper function of autophagy, the exact roles of factors responsible for positioning autophagosomes remain incompletely understood. In this study, we performed a loss-of-function genetic screen targeting genes potentially involved in microtubular motility. A genetic background that blocks autophagosome-lysosome fusions was used to accurately analyze autophagosome positioning. We discovered that pre-fusion autophagosomes move towards the non-centrosomal microtubule organizing center (ncMTOC) in *Drosophila* fat cells, which requires a dynein-dynactin complex. This process is regulated by the small GTPases Rab7 and Rab39 together with their adaptors: Epg5 and ema, respectively. The dynein-dependent movement of vesicles toward the nucleus/ncMTOC is essential for efficient autophagosomal fusions with lysosomes and subsequent degradation. Remarkably, altering the balance of kinesin and dynein motors changes the direction of autophagosome movement, indicating a competitive relationship where normally dynein-mediated transport prevails. Since pre-fusion lysosomes were positioned similarly to autophagosomes, it indicates that pre-fusion autophagosomes and lysosomes converge at the ncMTOC, which increases the efficiency of vesicle fusions.

## Introduction

Macroautophagy is an essential self-degradation pathway in eukaryotic cells, during which double-membrane-bound autophagosomes transport materials to lysosomes for degradation (*Parzych and Klionsky, 2014*). Defects in autophagy are associated with multiple pathologies, prompting extensive study of its molecular players over the past decades (*Lei and Klionsky, 2021*). During

macroautophagy, a double-membrane structure called an autophagosome is formed in an *atg* gene-dependent manner. This autophagosome then fuses with a lysosome or late endosome in a process dependent on the small GTPases Rab7, Rab2, and Arl8, the HOPS tethering complex, and the SNARE proteins Syntaxin17, Ubisnap/SNAP29, and Vamp7 (*Lőrincz and Juhász, 2020*).

Autophagosomes are suggested to form at random locations within the cytoplasm and are subsequently transported toward the cell center (*Jahreiss et al., 2008*; *Kimura et al., 2008*). Establishing proper proximity between autophagosomes and lysosomes is essential for their ability to fuse. The microtubular network and associated motor proteins are crucial for most vesicular transport. Thus, the involvement of the microtubular system has been suggested in various aspects of autophagosome dynamics, including biogenesis, transport, amphisome formation (*Köchl et al., 2006*), and autophagic clearance (*Ravikumar et al., 2005*). It is proposed that while microtubules are necessary for the maturation of autophagosomes, their fusion capacity is independent of microtubules (*Fass et al., 2006*). Nevertheless, dynein-regulated autophagosomal motility appears indispensable for efficient lysosomal fusion (*Jahreiss et al., 2008*; *Kimura et al., 2008*). How autophagosomes gain the ability to move along microtubules remains unclear. It is suggested that autophagic vesicles acquire dyneins by endosomal fusion (*Cheng et al., 2015*); however, autophagosomes still appear to be motile upon Syx17 loss (*Neisch et al., 2017*).

Most of our knowledge about autophagosome positioning and movement comes from studies on neurons, where autophagosomes form in the terminal part of axons and then travel toward the soma by dynein-dynactin-regulated bulk retrograde transport during basal autophagy (*Ikenaka et al., 2013*; *Lee et al., 2011*; *Maday et al., 2012*; *Wang et al., 2015*). During their route, they fuse with endosomes and lysosomes, resulting in gradual acidification and the acquisition of lysosomal markers (*Kargbo-Hill and Colón-Ramos, 2020*; *Lee et al., 2011*). Degradation takes place in the cell body (*Maday et al., 2012*). Inhibited retrograde transport leads to neurodegeneration and defective removal of synaptic autophagosomes (*Fu et al., 2014*; *Lee et al., 2011*; *Ravikumar et al., 2005*), highlighting the importance of axonal transport in the acidification and degradation processes.

Autophagosomes are suggested to use both dyneins and kinesins in neuronal cells. Initially, they exhibit bidirectional motility at the axon tip, later shifting to dynein-regulated retrograde transport directed toward the soma, where the already mature autolysosomes again show bidirectional motility (*Maday et al., 2012*). Various scaffolding proteins have been found to regulate autophagosome transport, including CKA as part of the STRIPAK complex (*Neisch et al., 2017*), JIP1, JIP3, and JIP4 (*Cason et al., 2021*; *Cason and Holzbaur, 2023*; *Fu et al., 2014*), as well as Huntingtin and HAP1 (*Cason et al., 2021*; *Kargbo-Hill and Colón-Ramos, 2020*; *Wong and Holzbaur, 2014*). It is important to note that before autophagosome closure, phagophores are not able to be transported (*Fass et al., 2006*).

However, most of our knowledge about autophagosome motility comes from experimental methods and tools that do not distinguish between non-acidic autophagosomes and autophagic structures that have already undergone some endolysosomal fusion and acidification. For example, several studies used reporters such as LC3 fused to red fluorescent proteins, which also label post-fusion autolysosomes. Consequently, autophagic organelles that have acidified are sometimes still considered autophagosomes, when in fact they could be autolysosomes.

We and our colleagues have previously identified and characterized several key players in autophagosome-lysosome fusion in starved *Drosophila* fat cells (*Boda et al., 2019*; *Hegedűs et al., 2016*; *Lőrincz et al., 2017b*; *Takáts et al., 2013*; *Takáts et al., 2014*). During these studies, we observed an intriguing phenomenon: despite being generated at random positions in the cytosol, autophagosomes accumulated around the nucleus when either the HOPS tethering complex or the SNARE fusion machinery was inhibited (*Boda et al., 2019*; *Lőrincz et al., 2019*; *Takáts et al., 2013*; *Takáts et al., 2014*).

This observation led us to employ a novel and unique approach by examining autophagosome positioning in cells where autophagosome-lysosome fusion was inhibited using HOPS RNAi. This method allowed us to exclude the confounding effects of ongoing vesicle fusions, which could otherwise obscure the accurate determination of the roles of different factors in vesicle positioning. These cells were utilized in an RNA interference screen to identify and characterize the molecular participants involved in autophagosome positioning. Our work represents the first comprehensive description of

the transport machinery involved in pre-fusion autophagosomes and its significance during autolysosome formation.

## Results

We began our investigations by performing an RNA interference screen to identify genes potentially involved in the microtubular positioning of autophagosomes, including MT proteins and motors (such as dynein, dynactin, and kinesin subunits), Rab small GTPases, and their effectors (for the complete list of tested genes and results, see *Supplementary files 1 and 2*). We used larval fat cells of the fruit fly (*Drosophila melanogaster*) as a model system, in which bulk macroautophagy was induced by starvation (*Scott et al., 2004*). The fat tissue contained GFP-positive mosaic cells, in which we silenced the gene of interest together with the Vps16A central subunit of the HOPS complex. This RNAi effectively impairs autophagosomal fusion, leading to the accumulation of autophagosomes (*Takáts et al., 2014*). Fat cells also expressed an mCherry-Atg8a reporter driven by a *UAS*-independent fat body-specific *R4* promoter. This reporter marks both autophagosomes and autolysosomes in control cells, due to the stability of mCherry in acidic environments (*Figure 1A*). However, in *vps16a* RNAi cells, which also expressed a control (*luciferase*) RNAi, we observed the accumulation of small mCherry-Atg8a puncta—representing autophagosomes—in the perinuclear region (*Figure 1A, B and F*), consistent with previous observations (*Takáts et al., 2014*).

Similar observations were made by endogenous immunostaining against Atg8a (*Figure 1—figure supplement 1A, H*), Rab7 (*Figure 1—figure supplement 1B, I*), and Arl8 (*Figure 1—figure supplement 1C, J*). Endogenous Atg8a immunostaining is specific for autophagosomes (*Lőrincz et al., 2017a*), Rab7 antibody labels late endosomes, lysosomes, and autophagosomes (*Hegedűs et al., 2016*), while Arl8 is a lysosome-specific small GTPase responsible for lysosomal motility and autophagosome-lysosome fusion (*Bagshaw et al., 2006*; *Boda et al., 2019*; *Hofmann and Munro, 2006*). Our results thus indicate that cells with impaired autophagosome-lysosome fusion accumulate not only pre-fusion autophagosomes but also pre-fusion lysosomes or late endosomes around their nuclei. Although Atg8a signal intensity detected by immunohistochemistry is higher in fusion-incompetent cells than in adjacent control cells due to the accumulation of non-degraded Atg8a (*Figure 1—figure supplement 1A*), the signal of the mCherry-Atg8a reporter appears slightly weaker in *vps16a* RNAi cells compared to controls (*Figure 1B*). This is because the reporter is highly stable and resistant to the acidic environment in autolysosomes (*Lőrincz et al., 2017b*). To confirm that our mCherry-Atg8a reporter labels structures of autophagic origin, we co-expressed an *atg8a* RNAi with *vps16a* RNAi, which effectively removed the signal of mCherry-Atg8a from the mosaic cells (*Figure 1—figure supplement 1D, K*), confirming that this reporter does not label non-autophagic vacuoles in these cells. Furthermore, the mCherry-Atg8a signal was almost completely absent in *atg1*; *vps16a* double RNAi cells (*Figure 1—figure supplement 1E, K*), indicating that this reporter is transported to autolysosomes via autophagosomes. To exclude the possibility that the perinuclear accumulation of autophagosomes and unfused lysosomes is due to the overall disorganization of organelles in Vps16A-depleted cells, we immunostained the cells against Gmap, which revealed that the positions of the Golgi apparatuses remained similar to control upon Vps16A silencing. Additionally, the fluorescent signal of Gmap was increased in Vps16A-depleted cells, which is consistent with the fact that the Golgi apparatus is a substrate of golgiphagy in flies (*Rahman et al., 2022*; *Figure 1C, G and H*).

### The transport of autophagosomes is microtubule-dependent and minus-end directed

As microtubule (MT)-associated autophagosome transport has been suggested to be more prominent compared to the actomyosin network (*Lőrincz and Juhász, 2020*), we first silenced the microtubule subunits α- and β-tubulin in *vps16a* RNAi cells to clarify whether the perinuclear localization of autophagosomes is indeed established by the MT network. As expected, knockdown of tubulins diminished the perinuclear localization of autophagosomes and led to their scattering in the cytoplasm (*Figure 1D, E, I*). Larval fat body cells have been shown to possess a perinuclear, non-centrosomal MTOC (ncMTOC) (*Zheng et al., 2020*). This ncMTOC is stabilized by the Spectraplakin Short stop (Shot), and its depletion translocates the perinuclear ncMTOC to an ectopic, cytosolic location (*Sun et al., 2019*; *Zheng et al., 2020*). Therefore, we hypothesized that autophagosomes travel towards

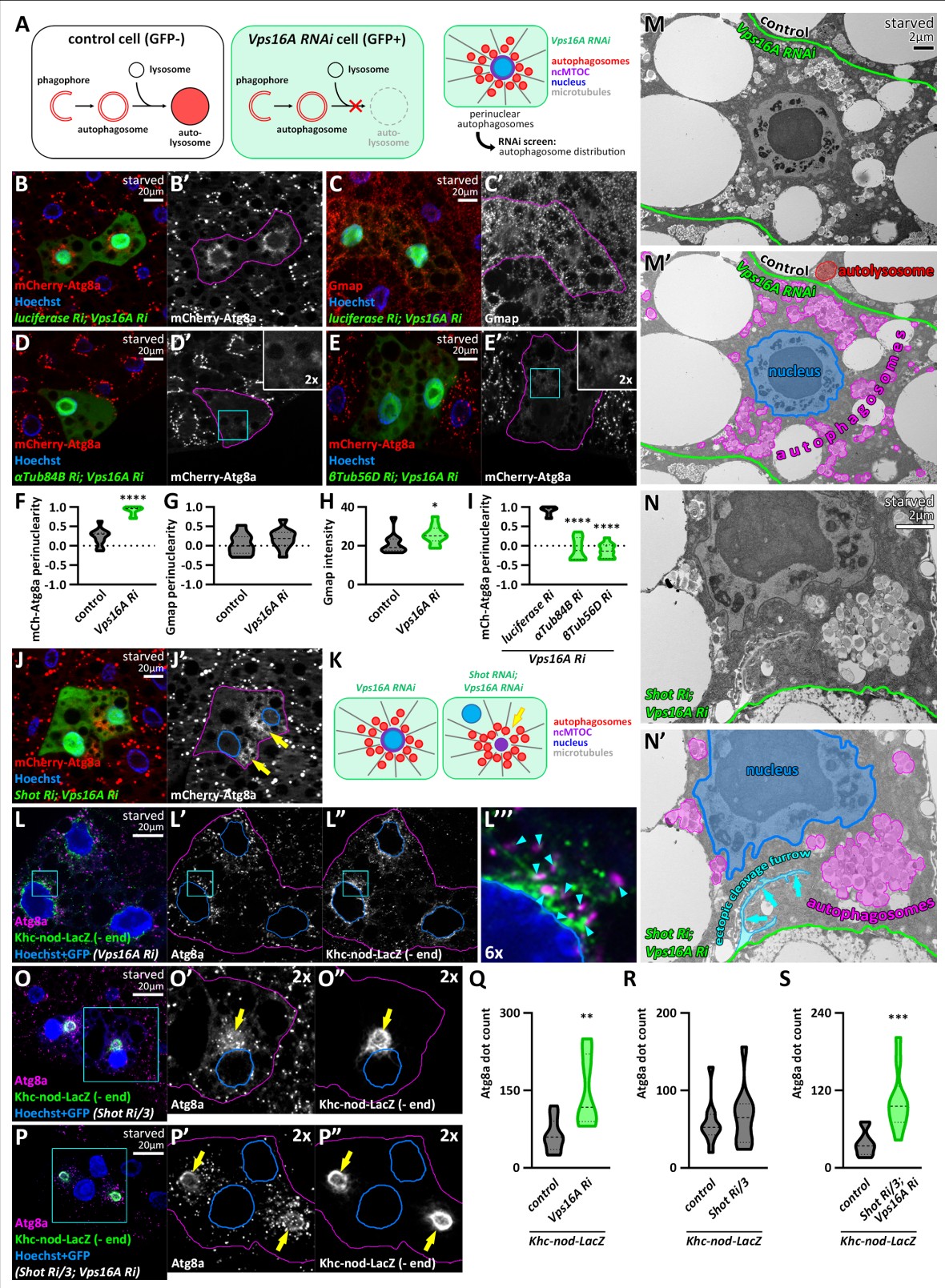

**Figure 1.** Autophagosomes move towards the non-centrosomal microtubule organizing center (ncMTOC) in fat cells. (**A**) Schematic drawing of the experimental design for screening. (**B**) Non-fused autophagosomes accumulate in the perinuclear region upon Vps16A silencing. (**C**) The cis-Golgi compartment remains unchanged upon the expression of *vps16a* RNAi. (**D, E**) The accumulation of autophagosomes is not perinuclear in α or β *tubulin; vps16a* double RNAi cells. The boxed areas in the main panels, marked by cyan, are enlarged in the insets (**D′ and E′**). (**F–I**) Quantification of data

*Figure 1 continued on next page*

*Figure 1 continued*

shown in B-E; n=10 cells. (**J, K**) Autophagosomes position at an ectopic MTOC (yellow arrows) formed upon Shot silencing. (29 out of 41 cells exhibited an ectopic MTOC = 70.73%). (**L**) Autophagosomes are near microtubule minus-ends, marked by Khc-nod-LacZ. The boxed area in the main panels, marked by cyan, is enlarged in **L'''** (proximity sites indicated by cyan arrowheads). (**M, N**) Correlative ultrastructural analysis shows autophagosomes accumulating near the nucleus upon Vps16A silencing (border of control and silencing cells marked by green) (**M**). Shot knockdown causes aggregation of autophagosomes in ectopic foci in *vps16a* RNAi cells (**N**). An ectopic cleavage furrow (hallmark of Shot depletion **Sun et al., 2019**) is also visible (cyan arrows in **N'**). Note: the magnification of N is higher to better show this structure. (**O, P**) Atg8a positive autophagosomes are clustered around the ectopic MTOC (yellow arrows) which is encircled by the signal of the minus-end marker Khc-nod-LacZ in *shot* RNAi (**O**) and *vps16a; shot* double RNAi cells (**P**). The boxed areas in the main panels, marked by cyan, are enlarged in **O', O'', P', and P''**. Nuclei are outlined in blue in **J', L'-L''', O', O'', P', and P''**. The GFP signal of RNAi and Khc-nod-LacZ expressing cells is false-colored blue in composite images. (**Q–S**) Quantification of data shown in L, O, P; n=10 cells. The boundaries of RNAi cells are highlighted in magenta in the grayscale panels.

The online version of this article includes the following figure supplement(s) for figure 1:

**Figure supplement 1.** Additional data on ncMTOC-oriented autophagosome transport in fat cells.

this MTOC in starved fat cells, and it is the position of the ncMTOC, rather than the nucleus, that determines their direction. Accordingly, autophagosomes accumulated in a central cytosolic region rather than around the nucleus in *shot, vps16a* double RNAi cells (**Figure 1J and K**, **Figure 1—figure supplement 1F, G**).

To further confirm that autophagosomes indeed travel towards the ncMTOC, we expressed the MT minus-end reporter Khc-nod-LacZ (a hybrid recombinant kinesin) (**Clark et al., 1997**) in *vps16a* RNAi cells. Immunolabeling of Atg8a-marked autophagosomes revealed their close proximity to the reporter, which effectively labeled the perinuclear MT network (**Figure 1L**). Additionally, we performed ultrastructural analysis to further support our findings. Compared to the mostly perinuclear distribution of autophagosomes in Vps16A single knockdown cells (**Figure 1M**), s*hot, vps16a* double RNAi resulted in the concentration of autophagosomes in large groups adjacent to the nucleus, consistent with our fluorescent data (**Figure 1N**). Moreover, the groups of Atg8a-positive autophagic structures observed upon Shot depletion accumulate around a Khc-nod-LacZ-positive region, independently of Vps16A (**Figure 1—figure supplement 1O–S**). Taken together, our results indicate that autophagosomes move along the MT network oriented towards the MT minus-end to their final destination near the ncMTOC.

## A cytoplasmic dynein-dynactin complex transports autophagosomes

Next, we turned to microtubular motor complex subunits. Dynein complexes consist of motor domain-containing heavy chains (HC), intermediate chains (IC), light intermediate chains (LIC), and light chains (LC), and their functions are regulated by dynactin complexes (**Canty et al., 2021**; **Vaughan and Vallee, 1995**). The fruit fly genome contains two genes encoding HCs and LICs, one single IC gene, and several genes encoding LCs. These can form several cytoplasmic dynein complexes; our goal was to find the one(s) responsible for autophagosome transport. Upon silencing Dynein heavy chain 64 C (Dhc64C, a HC subunit), short wing (sw, an IC subunit), Dynein light intermediate chain (Dlic, a LIC subunit), and roadblock (robl, a LC subunit) in *vps16a*-silenced cells, we observed an interesting phenomenon: autophagosomes accumulated in the cell periphery, under the plasma membrane, and not around the nucleus (**Figure 2A–D and K**, **Figure 2—figure supplement 1A, R**). Similar to the dynein hits, the silencing of DCTN1-p150 (Dynactin 1, p150Glued homolog), DCTN2-p50, and DCTN4-p62 also resulted in the redistribution of autophagosomes to the cell periphery (**Figure 2E–G and K**, **Figure 2—figure supplement 1B, C, R**). Our mCherry-Atg8a data were strengthened by endogenous Atg8a and Rab7 immunostainings (**Figure 2—figure supplement 1D–I,T, U**), as Atg8a and Rab7 positive puncta also redistributed to the cell periphery upon co-silencing Vps16A with our dynein or dynactin hits.

We also overexpressed a dominant-negative form of DCTN1-p150, as well as a wild-type DCTN2-p50/dynamitin, which is described to cause dominant-negative effects when overexpressed (**Zheng et al., 2020**). These reproduced the phenotypes of dynein or dynactin loss, further strengthening our data (**Figure 2—figure supplement 1J, K, R**). In turn, silencing other dynein or dynactin genes did not cause similar effects; these cells were either control-like (*vps16a* single RNAi) (**Supplementary file 1**, **Supplementary file 2**) or, in two cases, a mild scattering of autophagosomes were observed (Dlc90F,

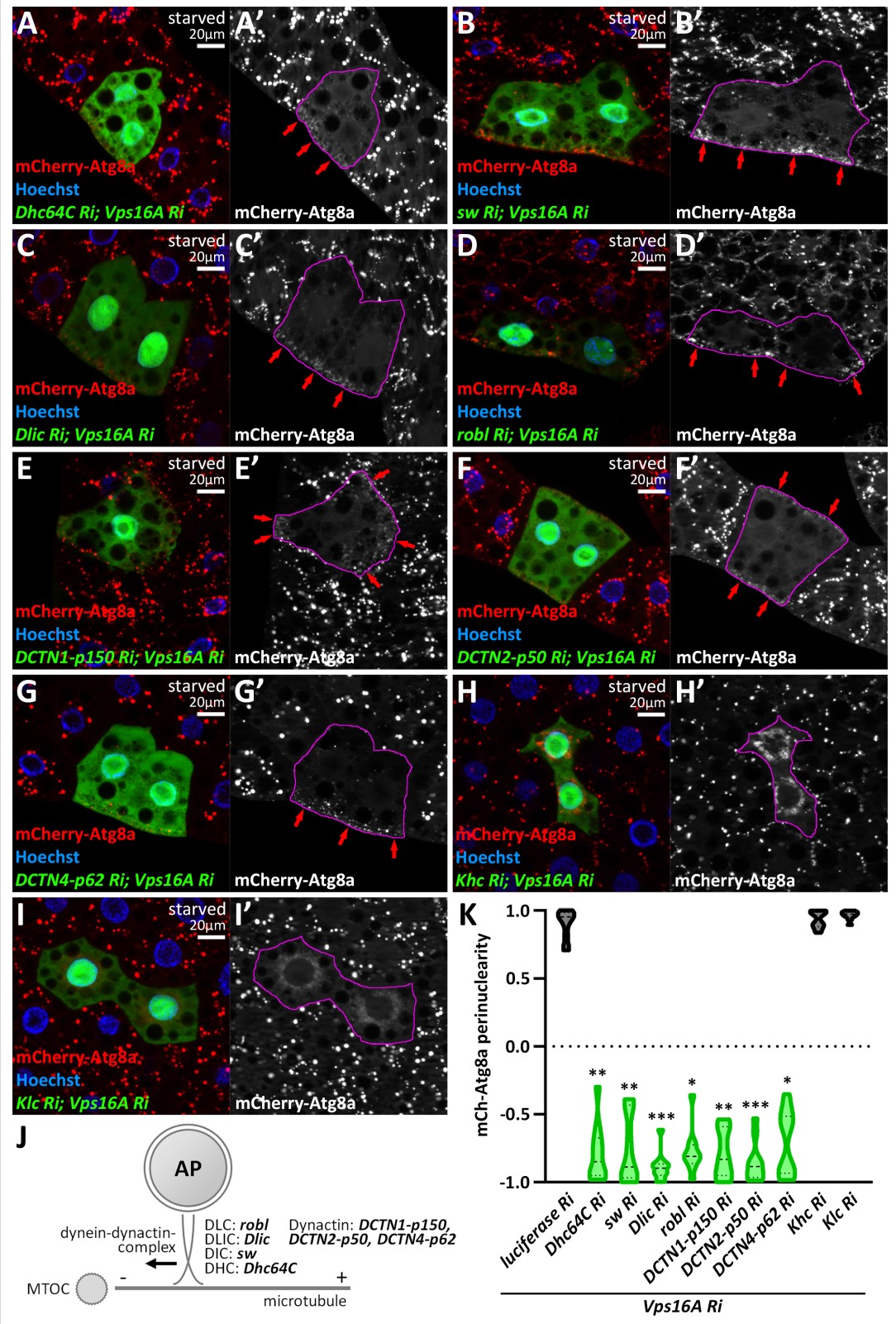

**Figure 2.** A dynein-dynactin complex is required for minus-end-directed autophagosome transport. (**A–G**) Knockdown of dynein (**A–D**) and dynactin subunits (**E–G**) results in the peripheral redistribution of autophagosomes in *vps16a* RNAi cells (red arrows). (**H, I**) Kinesin silencing does not affect the perinuclear accumulation of autophagosomes in *vps16a* RNAi cells. (**J**) Proposed model of the suggested dynein-dynactin complex responsible for

*Figure 2 continued on next page*

Figure 2 continued

autophagosome positioning in fat cells. DHC: dynein heavy chain; DIC: dynein intermediate chain; DLIC: dynein light intermediate chain; DLC: dynein light chain. (**K**) Quantification of data shown in **A-I**; n=10 cells. The boundaries of RNAi cells are highlighted in magenta in the grayscale panels.

The online version of this article includes the following figure supplement(s) for figure 2:

**Figure supplement 1.** Additional data on dynein regulated autophagosome transport.

an LC subunit, and capping protein alpha, cpa, a dynactin subunit) (*Figure 2—figure supplement 1L, M, S*, *Supplementary file 1*, *Supplementary file 2*).

Our results thus suggest that autophagosomes are mainly transported by a cytosolic dynein complex composed of Dhc64 (HC), sw (IC), Dlic (LIC), and roadblock (LC), regulated by a DCTN1-2-4 containing dynactin complex (*Figure 2J*). However, we cannot exclude the possibility that other dynein and dynactin subunits also contribute to autophagosome motility, as this would require confirmation that the RNAi transgenes yielding negative results were indeed efficient in generating loss-of-function of their target genes. Importantly, the peripheral accumulation of autophagosomes upon the lack of dynein-regulated movement suggests that kinesins can take their place and carry autophagosomes to the positive end of microtubules.

We continued screening by silencing dynein activators and regulators. These proteins, including the Bicaudal-D, Hook, and Ninein families in mammals, are responsible for enhancing processive motility and recruiting cargo to the dynein-dynactin complex (*Olenick and Holzbaur, 2019*; *Redwine et al., 2017*). We found that the silencing of a candidate activator, Girdin (*Redwine et al., 2017*), in a *vps16a* RNAi background led to the dispersal of autophagosomes (*Figure 2—figure supplement 1N, V*), similarly to the loss of Lis-1, a well-studied and essential regulator of dynein motor function (*Dix et al., 2013*; *Siller et al., 2005*; *Sitaram et al., 2012*; *Swan et al., 1999*; *Figure 2—figure supplement 1O, V*). This can be explained by the fact that some dynein function is still present without Girdin and Lis-1, and their loss does not completely abolish dynein activity.

Therefore, next, our screen focused on kinesin motors. Importantly, none of the kinesin knockdowns inhibited or significantly enhanced the perinuclear positioning of autophagosomes in *vps16a*-depleted cells (Khc and Klc as examples are shown, *Figure 2H, I and K*, *Figure 2—figure supplement 1P, Q, T*, *Supplementary file 1*, *Supplementary file 2*). These results suggest that cells predominantly use the dynein complex to transport autophagosomes, rather than kinesins.

## A proper dynein/kinesin ratio determines the direction of autophagosome positioning

Since dynein loss leads to the peripheral relocation of autophagosomes, we hypothesized that in this case, kinesins take over the role of transporting autophagosomes, preferring the opposite direction. To examine this possibility, we overexpressed two kinesin motors (Klp67A, Klp98A) in *vps16a* RNAi cells. Strikingly, both resulted in the scattering of autophagosomes, and in some cases, caused the peripheral accumulation of mCherry-Atg8a puncta, resembling dynein loss (*Figure 3A, B and F*). Moreover, autophagosomes were scattered upon co-silencing a dynactin and a kinesin in *vps16a* RNAi cells (*Figure 3C and F*). This suggests that without MT motors, autophagosomes are unable to move properly.

To further examine the relationship between dyneins and kinesins, we expressed the recombinant kinesin Khc-nod-LacZ in *vps16a* single and dynactin, *vps16a* double RNAi cells. This recombinant kinesin contains the cargo domain of Khc but moves towards the MT minus-end, similar to dyneins. Interestingly, overexpression of Khc-nod-LacZ in *vps16a* single RNAi cells appeared to influence the assembly and/or function of the ncMTOC, as the nuclear envelope was only partly surrounded by autophagosomes, but their distribution remained perinuclear (*Figure 3D and F*). Strikingly, when we overexpressed this recombinant kinesin in dynactin, Vps16A double knockdown cells, autophagosomes became dispersed (*Figure 3E and F*), and no longer accumulated at the periphery as seen in *dynactin1-p150*, *vps16a* double RNAi cells. Since kinesins do not require the activity of dynactins, this suggests that this recombinant kinesin could partially rescue the dynein/dynactin function loss and take over the role of the missing minus-end motors, further supporting that a proper dynein/kinesin ratio determines the direction of autophagosome positioning.

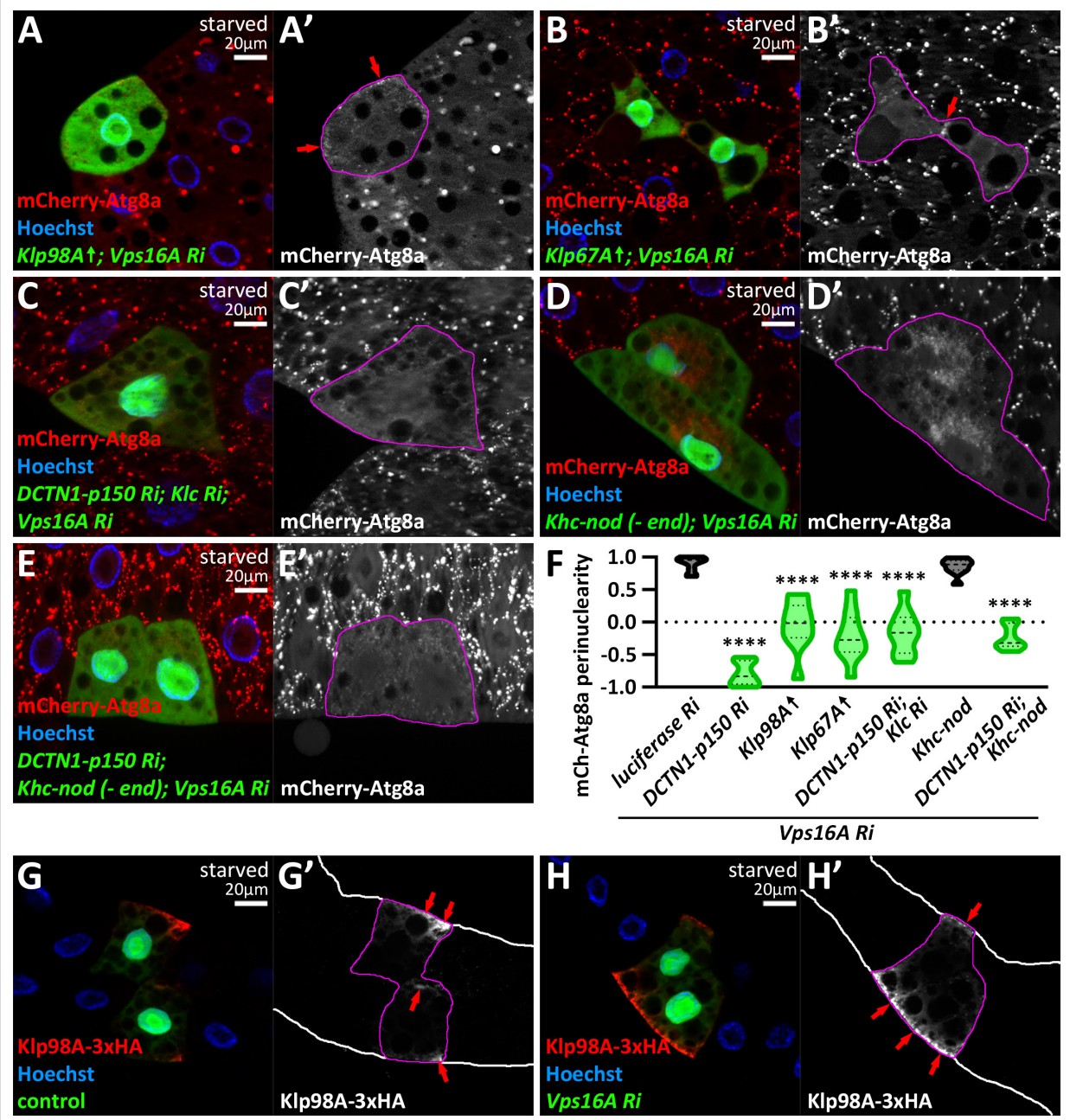

**Figure 3.** The proper dynein/kinesin ratio determines the directionality of autophagosome transport. (**A, B**) Overexpression of kinesin motors blocks the minus-end transport of autophagosomes, leading to their accumulation in the cell periphery in *vps16a* RNAi cells (red arrows). (**C**) In dynactin-kinesin-*vps16a* triple RNAi cells, autophagosomes are distributed throughout the cytoplasm. (**D**) Overexpression of the recombinant minus-end motor Khc-nod-LacZ does not affect minus-end directed autophagosome transport in *vps16a* RNAi cells. (**E**) Khc-nod-LacZ expression partially rescues the dynactin KD-induced peripheral redistribution of autophagosomes in *vps16a* RNAi cells. (**F**) Quantification of data shown in **A-E**; n=10. Data for *dctn1-p150* RNAi is included as a positive control for peripheral distribution (shown in *Figure 2E, K*). (**G, H**) Overexpressed Klp98A-3xHA accumulates at the periphery of marginal fat body cells, specifically on the side facing the body cavity and in contact with the hemolymph (red arrows), in both control (**G**) and fusion-inhibited (*vps16a* RNAi) cells (**H**). The boundaries of RNAi or kinesin overexpressing cells are highlighted in magenta in the grayscale panels. Fat body edges are outlined in white in (**G′ and H′**).

Since peripheral autophagosome accumulation upon loss of dyneins and dynactins tended to occur on the side of marginal fat body cells facing the body cavity and in contact with the blood (*Figure 2A–G*), we asked whether kinesins also follow a similar distribution pattern. To examine this, we performed anti-HA immunostaining on HA-tagged Klp98A-overexpressing cells and found that

this reporter is indeed enriched at the fat body margin (*Figure 3G and H*), supporting the hypothesis that the autophagosome redistribution upon loss of minus end transport is kinesin-dependent.

## Rab7 and Rab39 and their effectors Epg5 and ema are required for bidirectional autophagosome transport

Rab small GTPases regulate vesicle transport and fusion by recruiting different effectors in their active, GTP-bound form (*Stenmark, 2009*). Therefore, we screened all the Rab small GTPases. In Rab7, or subunits of its guanine nucleotide exchange factor complex (Mon1-Ccz1) and its interactor Epg5 (*Gillingham et al., 2014*) knockdown cells with simultaneous Vps16A silencing, we could no longer observe the perinuclear accumulation of autophagosomes; they were scattered throughout the cytoplasm (*Figure 4A–D and J*). Other hits were Rab39 and its interactor ema (*Gillingham et al., 2014*), both suggested to be involved in the regulation of lysosomal degradation (*Kim et al., 2012*; *Kim et al., 2010*; *Lakatos et al., 2021*; *Zhang et al., 2023*). Their phenotype was very similar (*Figure 4E, F and J*, *Figure 4—figure supplement 1A, G*) to the loss of Rab7 and Epg5.

Importantly, no other interaction partners of Rab7 and Rab39 appeared to be required for autophagosome transport (*Supplementary file 1*, *Supplementary file 2*; examples shown in *Figure 4G, H and J*), suggesting that mainly the Rab7-Epg5 and Rab39-ema interactions are required for the bidirectional motility of autophagosomes (*Figure 4I*). Our results were also strengthened by immunolabelings of Atg8a and Rab7 (*Figure 5A, B, E and F*, *Figure 4—figure supplement 1B–F, H, I*). Importantly, autophagosomes were still dispersed in *epg5*, *vps16a* double RNAi cells, even if we overexpressed a YFP-Rab7 transgene, suggesting that Rab7 indeed regulates autophagosome positioning via Epg5 (*Figure 5C and G*). Interestingly, Arl8 immunolabeling revealed that Epg5 loss does not influence the perinuclear positioning of non-fused lysosomes (*Figure 5D and H*), suggesting that it exerts its function on autophagosomes.

Therefore, we further analyzed Epg5 functions. We first generated an *Epg5-9xHA* transgene driven by the *epg5* genomic promoter and expressed this reporter in *Drosophila* S2R+ cells. We found that this reporter colocalizes with both endogenous Rab7 and Atg8a (*Figure 5I and J*). Moreover, we observed Atg8a-positive structures that colocalized with Epg5-9xHA, but not with Lamp1-3xmCherry, as well as structures that were triple positive (*Figure 5K*), indicating that Epg5 is present on both autophagosomes and autolysosomes. Epg5 has been suggested as a Rab7 effector both in fly (*Gillingham et al., 2014*) and in mammalian cells (*Wang et al., 2016*), which we could confirm as Epg5-9xHA coprecipitates with Rab7-FLAG in cultured fly cells (*Figure 5L*). Moreover, Epg5-9xHA also coprecipitates with the endogenous dynein motor Dhc64C (*Figure 5M*), supporting the idea that Rab7 via Epg5 is required for dynein-dependent autophagosome transport. Taking into consideration that Epg5 was found to regulate the positioning of autophagosomes but not lysosomes, we utilized garland nephrocytes to study its effect on endolysosome maturation. Nephrocytes maintain a constant rate of endocytosis, making them ideal tools to study the endolysosomal system (*Lőrincz et al., 2016*). We have previously shown that inhibited late endosome to lysosome maturation leads to the enlargement of the late endosomal compartment (*Boda et al., 2019*; *Hargitai et al., 2025*; *Lőrincz et al., 2019*; *Lőrincz et al., 2016*; *Lőrincz et al., 2017b*). However, we found that neither the Rab7 or FYVE-GFP-positive endosomal nor the Lamp1-positive lysosomal compartment changed upon the expression of *epg5* RNAi in nephrocytes (*Figure 5—figure supplement 1A–G*). This was strengthened by ultrastructural analyses, which showed no obvious changes in the morphology of the endolysosomal compartment in *epg5* RNAi nephrocytes (*Figure 5—figure supplement 1H, I*). These results suggest that in flies, Epg5 functions primarily in the autophagic pathway, independently from the endosomal system.

Knockdown of other small GTPases that play essential roles in the lysosomal system, such as Rab2 (*Lőrincz et al., 2017b*), Rab5 (*Poteryaev et al., 2010*), Rab14 (*Mauvezin et al., 2016*), or Arl8 (*Boda et al., 2019*), did not change the perinuclear pattern of mCherry-Atg8a-positive autophagosomes in *vps16a* RNAi cells (*Figure 4—figure supplement 2A–D, G*). Notably, silencing the recycling endosomal Rab11 resulted in the scattering of mCherry-Atg8a puncta and the exceptionally strong accumulation of autophagosomes revealed by endogenous Atg8a staining (*Figure 4—figure supplement 2E–H*). Rab11 has been shown to be involved in autophagosome maturation in flies (*Szatmári et al., 2014*), but neither Rab11 interactors resulted in any change in the perinuclear autophagosome

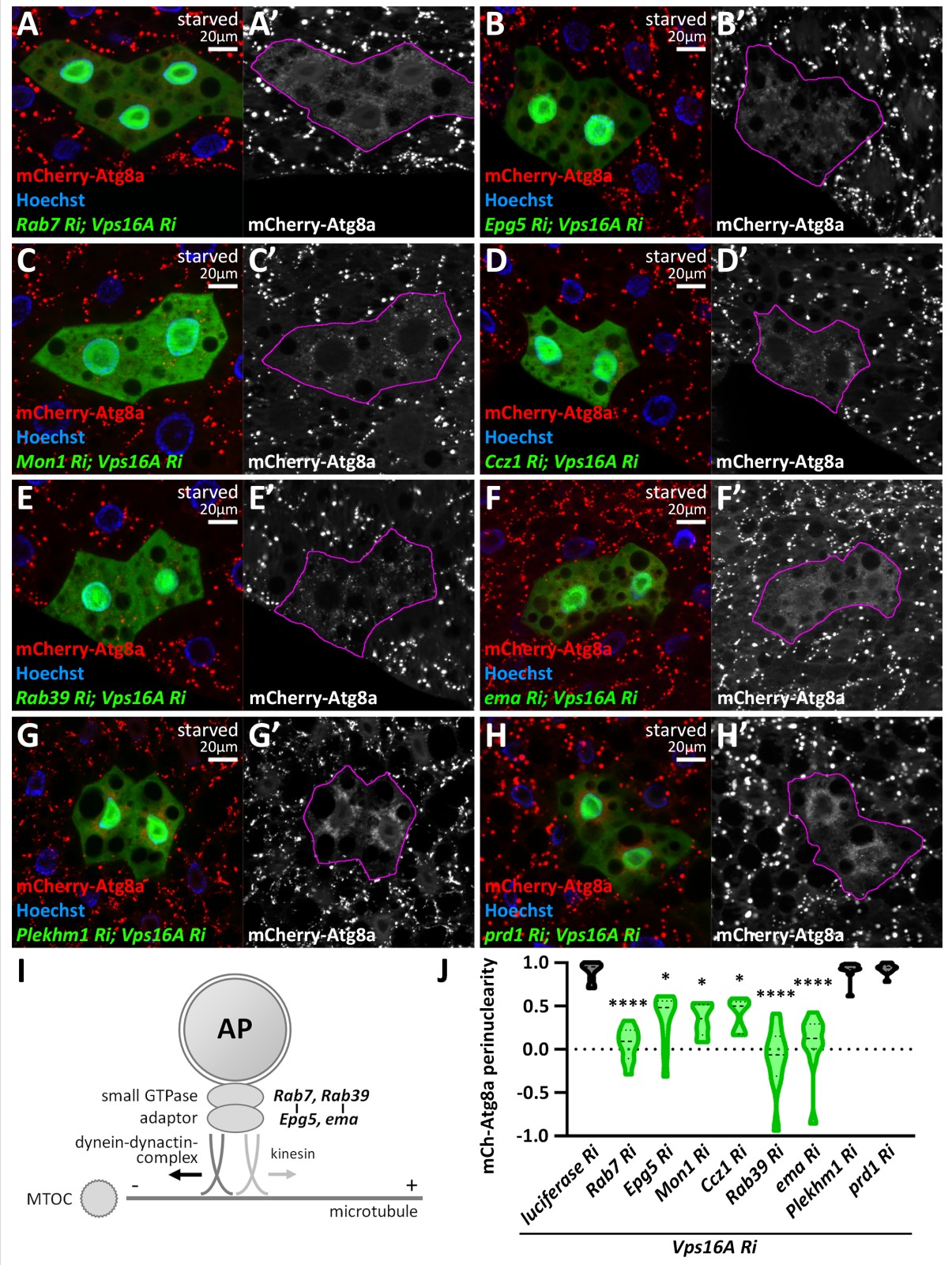

**Figure 4.** Rab7 and Rab39 small GTPases and their interactors are responsible for bidirectional movement of autophagosomes. (**A–H**) Knockdown of Rab7 (**A**), its interactor Epg5 (**B**), the subunits of its guanine nucleotide exchange factor Mon1 (**C**) and Ccz1 (**D**), as well as Rab39 (**E**) and its interactor ema (**F**), inhibits the perinuclear positioning of autophagosomes in *vps16a* RNAi cells. In contrast, other factors such as Plekhm1 (**G**) and prd1 (**H**) do

*Figure 4 continued on next page*

*Figure 4 continued*

not affect autophagosome positioning. (**I**) Proposed model of Rab small GTPases with their adaptors involved in autophagosome positioning. (**J**) Quantification of data shown in **A-H**; n=10 cells. The boundaries of RNAi cells are highlighted in magenta in the grayscale panels.

The online version of this article includes the following figure supplement(s) for figure 4:

**Figure supplement 1.** Additional data on the small GTPases that are required for autophagosome transport.

**Figure supplement 2.** The effect of other endolysosomal small GTPases on autophagosome positioning.

**Figure supplement 3.** Overexpression of endolysosomal small GTPases does not alter the perinuclear distribution of autophagosomes.

distribution (*Supplementary file 1*, *Supplementary file 2*), suggesting that the effects of Rab11 may be indirect, compared to our Rab7 and Rab39 hits.

Next, we analyzed the effect and localization of overexpressed autophagosomal and endolysosomal Rabs in *vps16a* RNAi cells. Neither the wild-type nor the constitutively active forms of the overexpressed Rabs changed the perinuclear distribution of mCherry-Atg8a-positive autophagosomes (*Figure 4—figure supplement 3A, B, D-L, N*). Importantly, overexpression of both forms of YFP-tagged Rab7 (*Figure 4—figure supplement 3A, B*) and Rab2 (*Figure 4—figure supplement 3D, E*), as well as endogenous Rab7 immunolabeling (*Figure 4—figure supplement 3C*) showed obvious autophagosomal localization. Moreover, wild-type YFP-Rab39 also overlapped with the mCherry-Atg8a puncta (*Figure 4—figure supplement 3F*) in *vps16a* RNAi cells. Our results suggest that Rab2 is exclusively required for autophagosomal fusions, while Rab7 and Rab39 are also required for autophagosome movement (*Figure 4—figure supplement 3M*). The YFP-tagged wild-type form of Rab14, which was described as a regulator of autophagic vesicle transport and fusion (*Mauvezin et al., 2016*), exhibited a punctate pattern, but did not localize to autophagosomes, suggesting that it localizes to other organelles, most likely lysosomes (*Figure 4—figure supplement 3L*). Taken together, Rab7 and Rab39, as well as their effectors, Epg5 and ema, respectively, appear to be the most important regulators responsible for microtubular autophagosome motility in both directions.

## Autophagosome transport machinery functions similarly in *snap29* and *vps16a* RNAi cells

While our findings strongly suggest that pre-fusion autophagosomes exhibit default minus- end–directed motility, we could not exclude the possibility that this phenotype specifically results from loss of the HOPS complex. To address this, we aimed to generate a *Drosophila* line with GFP-positive mosaic fat body cells expressing RNAi against one of the SNARE proteins required for autophagosome-lysosome fusion. Syntaxin 17 (Syx17), the autophagosomal SNARE (*Lőrincz and Juhász, 2020*; *Takáts et al., 2013*), appeared to be the obvious choice. However, in Syx17 knockdown nephrocytes, we observed a tethering lock that permanently anchors lysosomes and autophagosomes together (*Hargitai et al., 2025*), making it impossible to study autophagosome motility independently of lysosomes. Therefore, we turned to the SNARE Snap29 as an alternative and generated a *Drosophila* line with GFP-positive mosaic fat body cells expressing Snap29 RNAi. Like our vps16a RNAi line used in the genetic screen, these flies also expressed an mCherry-Atg8a reporter driven by a UAS-independent, fat body-specific R4 promoter. Snap29 encodes a SNARE protein essential for autophagosome–lysosome fusion (*Lőrincz and Juhász, 2020*; *Takáts et al., 2013*).

We crossed this line with a control (luciferase) RNAi and with several key hits from our screen. Similar to *vps16a* RNAi, Snap29 knockdown led to perinuclear accumulation of autophagosomes (*Figure 6A, I*). In *snap29* RNAi cells, *shot* and *dhc64c* RNAi redistributed autophagosomes to an ectopic MTOC or to the periphery, respectively (*Figure 6B, C, I*). Knockdown of the kinesin Khc did not affect the perinuclear accumulation of autophagosomes in *snap29* RNAi cells (*Figure 6D, I*), while RNAi targeting the small GTPases Rab7, Rab39, or their adaptors (Epg5 or ema) caused autophagosomes to become dispersed throughout the cytosol (*Figure 6E-I*).

These results indicate that the redistribution of autophagosomes toward the microtubule minus-end is a general consequence of impaired fusion, and further suggest that the autophagosome transport machinery functions independently of Vps16A and the HOPS complex.

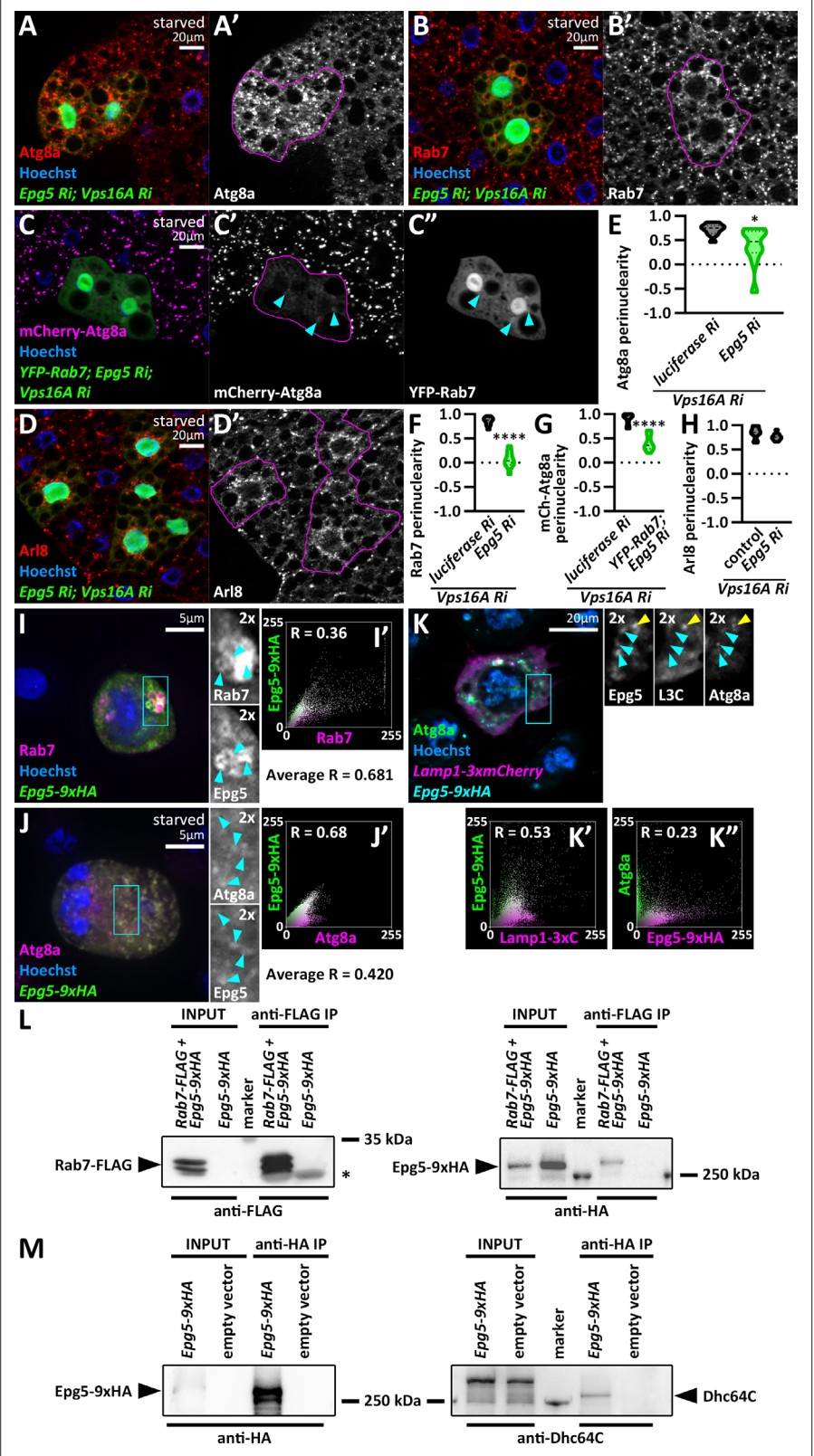

**Figure 5.** Epg5 is responsible for bidirectional movement of autophagosomes. (**A–B**) The distribution of Atg8a (**A**) or Rab7 (**B**) positive autophagosomes becomes dispersed upon the expression of *epg5* and *vps16a* RNAi. (**C**) Overexpression of YFP-tagged Rab7 does not rescue the scattered distribution of mCherry-Atg8a-positive autophagosomes in the absence of Epg5 in *vps16a* RNAi cells, even though the colocalization of YFP-Rab7 with

*Figure 5 continued on next page*

*Figure 5 continued*

mCherry-Atg8a remains unaffected. Cyan arrowheads in the grayscale panels point to YFP-Rab7 and mCherry-Atg8a double-positive dots. (**D**) The localization of Arl8-positive lysosomes remains perinuclear in *epg5, vps16a* double RNAi cells. The boundaries of RNAi and YFP-Rab7-expressing cells are highlighted in magenta in the grayscale panels. (**E–H**) Quantification of data shown in A-D; n=10 cells. (**I, J**) Epg5-9xHA colocalizes with endogenous Rab7 (**I**) or Atg8a (**J**) positive structures in S2R + cells. Cyan arrowheads within insets (marked by cyan boxes in panels **I** and **J**) point to Epg5-9xHA and Rab7 or Atg8a double-positive structures, respectively. (**I′**) and (**J′**) show scatter plots generated from the images of cells in panels **I** and **J**, respectively, depicting the intensity correlation profiles of Epg5-9xHA with Rab7 or Atg8a. Pearson correlation coefficients (**R**) are indicated, with the average R (n=10 cells) also shown, indicating colocalization in both cases. (**K**) Epg5-9xHA colocalizes with Atg8a-positive, Lamp1-3xmCherry-negative (pre-fusion) autophagosomes, as well as with Atg8a and Lamp1-3xmCherry double-positive autolysosomes in S2R+ cells. Cyan arrowheads in insets (marked by a cyan box in panel K) point to Epg5-9xHA, Atg8a double-positive, Lamp1-3xmCherry-negative structures, while a yellow arrowhead marks a triple positive autolysosome. (**K′ and K″**) Scatter plots based on the cell in panel K show intensity correlations of Epg5-9xHA with Lamp1-3xmCherry and Atg8a, respectively. Pearson correlation coefficients indicate partial colocalizations. (**L, M**) Coimmunoprecipitation experiments show that Epg5-9xHA binds to Rab7-FLAG (**L**) and endogenous Dhc64C (**M**) in cultured *Drosophila* cells. The asterisk in L marks immunoglobulin light chain. The smeared input bands of Dhc64C in panel M are due to the large size of Dhc64C, which affects its migration characteristics.

The online version of this article includes the following source data and figure supplement(s) for figure 5:

**Source data 1.** Zipped folder containing original files of the full raw uncropped, unedited blots for *Figure 5L and M*.

**Source data 2.** Zipped folder containing original files of the uncropped blots with the relevant bands clearly labeled for *Figure 5L and M*.

**Figure supplement 1.** Epg5 is not required for endosomal or lysosomal compartment integrity in garland nephrocytes.

## The transport of pre-fusion lysosomes is also minus-end directed

Given that *vps16a* RNAi led to the perinuclear distribution of immature lysosomes (*Figure 1—figure supplement 1C*) similar to autophagosomes, we hypothesized that pre-fusion autophagosomes and lysosomes travel in the same orientation, potentially sharing the same transport machinery. Thus, we stained lysosomes and co-expressed *vps16a* RNAi along with RNAi targeting our hits from the auto-phagosome positioning screen (*Figure 7*). Importantly, in most cases, we observed similar phenotypes with Lamp1 staining as we did with Atg8a: spectraplakin (*shot*) RNAi redistributed Lamp1 organelles to an ectopic ncMTOC, dynein inhibition redistributed lysosomes to the periphery, *rab7, rab39*, and *ema* RNAi-s resulted in the scattering of lysosomes across the cytosol, and kinesin depletion had no effect on the perinuclear accumulation of lysosomes (*Figure 7*).

However, there were two important exceptions: Epg5 knockdown left lysosomes perinuclear (*Figure 7C and L*), suggesting that it is indeed an autophagosomal adaptor. The other exception was *rab2* RNAi, which had no significant effect on autophagosome transport but resulted in dispersed, and sometimes even peripheral, lysosomal distribution in Vps16A-depleted cells (*Figure 7I and L*). This result suggests that Rab2 is a potential regulator of minus-end directed transport of lysosomes and that pre-fusion organelles (autophagosomes and lysosomes) predominantly travel towards the minus-end of the MTs and share the main molecular components.

## Minus-end-directed autophagosome and lysosome transport is required for autophagosome-lysosome fusion

We hypothesized that concentrating pre-fusion autophagosomes and lysosomes at the perinuclear region increases the probability of their fusion, thereby promoting autolysosome formation. To test this, we used similar reporters and reagents as above but did not silence Vps16A in the examined cells to allow autophagosome-lysosome fusions. First, we analyzed Epg5 knockdown, which has been described to inhibit autolysosome maturation based on GFP-Atg8a (*Byrne et al., 2016*). In line with this, 3xmCherry-Atg8a-positive autolysosomes were significantly smaller than in control cells, without any obvious change in their distribution (*Figure 8A, B, E and F*). Since 3xmCherry-Atg8a is transported to the lysosomes via autophagosome-lysosome fusion, and it retains its fluorescence

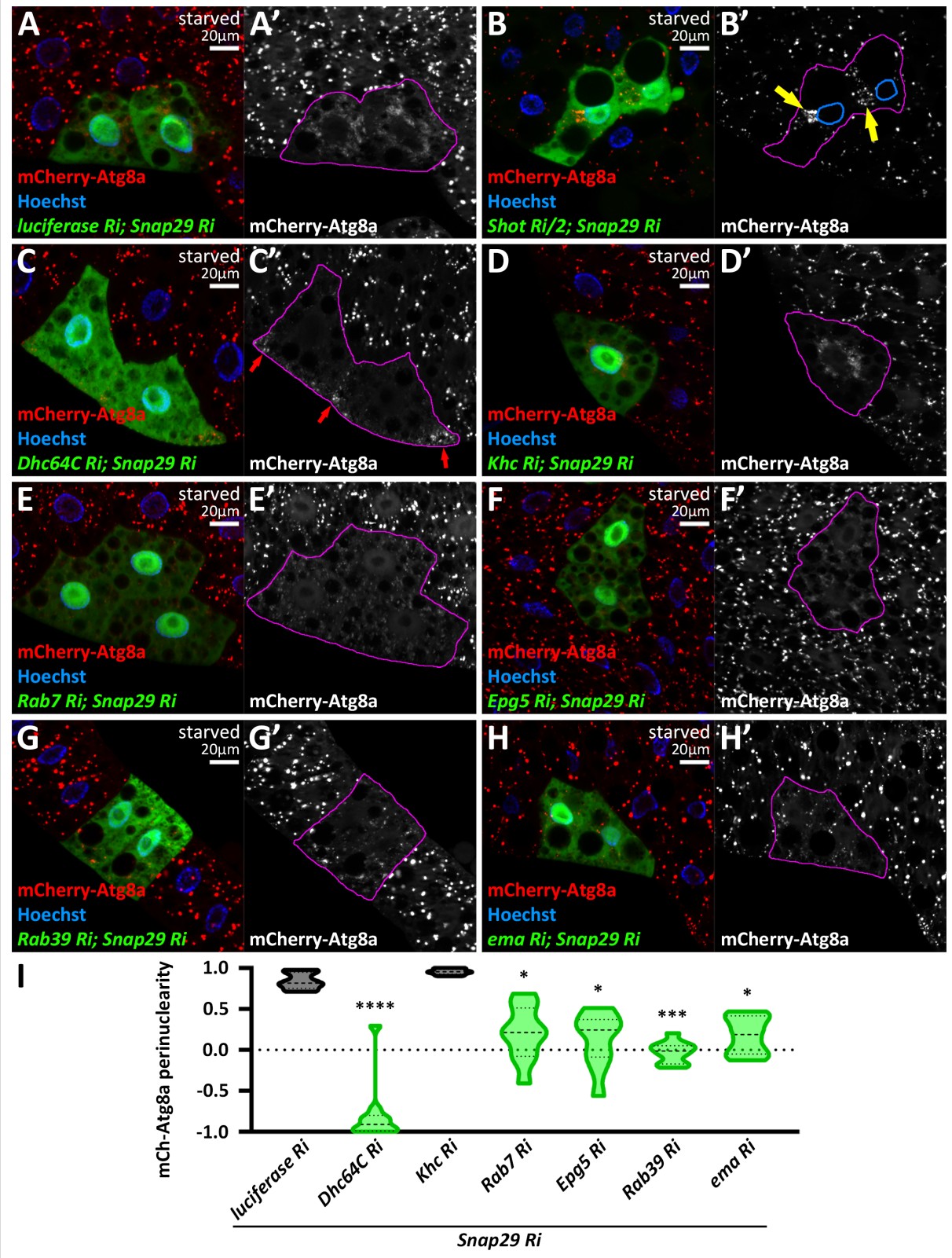

**Figure 6.** Knockdown of key regulators of autophagosome transport in a Snap29 RNAi background recapitulates the autophagosome distribution defects observed upon Vps16A KD. (**A–H**) Knockdown of key regulators of autophagosome transport in a Snap29 RNAi background results in autophagosome localization patterns similar to those observed with Vps16A RNAi. (**A–H**) In luciferase; Snap29 double knockdown cells, non-fused autophagosomes accumulate in the perinuclear region, marked by mCherry-Atg8a (**A**). Shot; Snap29 double knockdown causes autophagosomes to

*Figure 6 continued on next page*

*Figure 6 continued*

accumulate around an ectopic microtubule organizing center (MTOC) (**B**), marked by yellow arrows in **B′**. Nuclear outlines are shown in blue. Dhc64C knockdown in Snap29 RNAi cells causes autophagosomes to redistribute to the cell periphery (C, red arrows in **C′**). Khc knockdown does not alter the perinuclear distribution of autophagosomes seen in Snap29 RNAi cells (**D**). Co-knockdown of Snap29 with Rab7 (**E**), Epg5 (**F**), Rab39 (**G**), or ema (**H**) results in scattered autophagosome distribution throughout the cytoplasm. In grayscale panels, the boundaries of RNAi-expressing cells are highlighted in magenta. (**I**) Quantification of the data shown in panels (**A** and **C–H**). n=10 cells.

in the lysosomal environment, it can be used to monitor autolysosome maturation. Although we cannot determine the number of lysosomes that fuse with each autophagosome, the overall size of 3xmCherry-Atg8a-positive structures correlates with autolysosome maturation efficiency (*Lőrincz et al., 2017a*). Accordingly, the large Rab7 and Arl8 positive autolysosomes were almost completely absent from Epg5 silenced cells, as revealed by immunostainings (*Figure 8C, D, G, and H*), with only small autolysosomes present.

Silencing of the dynein and dynactin hits: Dhc64C, sw, Dlic, and robl, as well as DCTN1-p150, resulted in the redistribution of 3xmCherry-Atg8a-positive autolysosomes to the periphery and their size became significantly smaller (*Figure 9A–E, I and K*), suggesting that loss of minus-end directed transport leads to autolysosome maturation defects. In contrast, *khc* and *klc* RNAi caused the perinuclear accumulation of autolysosomes, which appeared larger than those in the controls, suggesting a trend, although this difference did not reach statistical significance (*Figure 9F, G, J and L*).

In accordance with its suggested role in autophagosome-lysosome fusion, silencing of Rab7 resulted in significantly smaller and mostly dispersed autolysosomes (*Figure 9—figure supplement 1A, D, E*; *Hegedűs et al., 2016*; *Lőrincz et al., 2017b*). Rab39 knockdown, however, led to mostly peripheral autolysosomes, which were not different in size from those in the control cells (*Figure 9—figure supplement 1B, D, E*), but the localization of the mCherry-Atg8a-positive autolysosomes suggests that Rab39 is also required for minus-end directed movement of mature lysosomes. Surprisingly, loss of ema did not influence the size and distribution of autolysosomes (*Figure 9—figure supplement 1C–E*).

Our most important finding came when we silenced the spectraplakin Shot, which caused the accumulation of autolysosomes in the ectopic cytosolic MTOC (*Figure 9H*). Notably, their size significantly increased (*Figure 9H and L*). This can be explained by the fact that the volume surrounding the ectopic ncMTOC in Shot-depleted cells is smaller than the volume around the nuclei, leading to an increased fusion rate in these cells. Taken together, these results demonstrate that minus-end transport is crucial for proper autolysosome maturation.

The observation that dispersing autophagosomes and lysosomes under the plasma membrane in dynein/dynactin-silenced cells leads to insufficient autolysosome maturation, while concentrating autophagosomes and lysosomes to an ectopic ncMTOC in *shot* RNAi results in the enlargement of lysosomes, indicates that the autophagosome-lysosome fusion rate depends on the volume of cytoplasm in which these organelles meet. Several studies have suggested a connection between microtubular transport and the fusion of autophagosomes (*Fass et al., 2006*; *Jahreiss et al., 2008*; *Kimura et al., 2008*; *Köchl et al., 2006*).

We tested this hypothesis by analyzing the colocalization between the 3xmCherry-Atg8a reporter and the lysosomal membrane protein Lamp1, either by immunostaining or by expressing GFP-Lamp1. Their overlap represents autolysosomes, while 3xmCherry-Atg8a-positive, Lamp1-negative structures are considered pre-fusion autophagosomes. Conversely, Lamp1-positive, 3xmCherry-Atg8a-negative structures indicate lysosomes of non-autophagic origin. Thus, reduced colocalization indicates a fusion defect. In control cells, Lamp1 or GFP-Lamp1 overlaps with mCherry-Atg8a, indicating that these organelles are indeed autolysosomes (*Figure 10A and H*, *Figure 10—figure supplement 1A, G*). In shot-depleted cells, both signals overlapped in the ectopic ncMTOC, indicating that autophagosomes can effectively fuse with lysosomes in this region (*Figure 10B and H*, *Figure 10—figure supplement 1B, G*). Accordingly, the loss of the kinesin heavy chain Khc did not alter the overlap of these markers, proving that autolysosomes could still be formed (*Figure 10C and H*, *Figure 10—figure supplement 1C, G*). Importantly, the loss of the dynein motor Dhc64C, as well as the dynactin subunit DCTN1-p150, greatly reduced the overlap of signals, indicating a less effective autophagosome-lysosome fusion (*Figure 10D, E and H*, *Figure 10—figure supplement 1D, G*, G). Since Rab7 and Epg5 have been implicated in autolysosome maturation, their depletion reduced the overlap of autophagic and

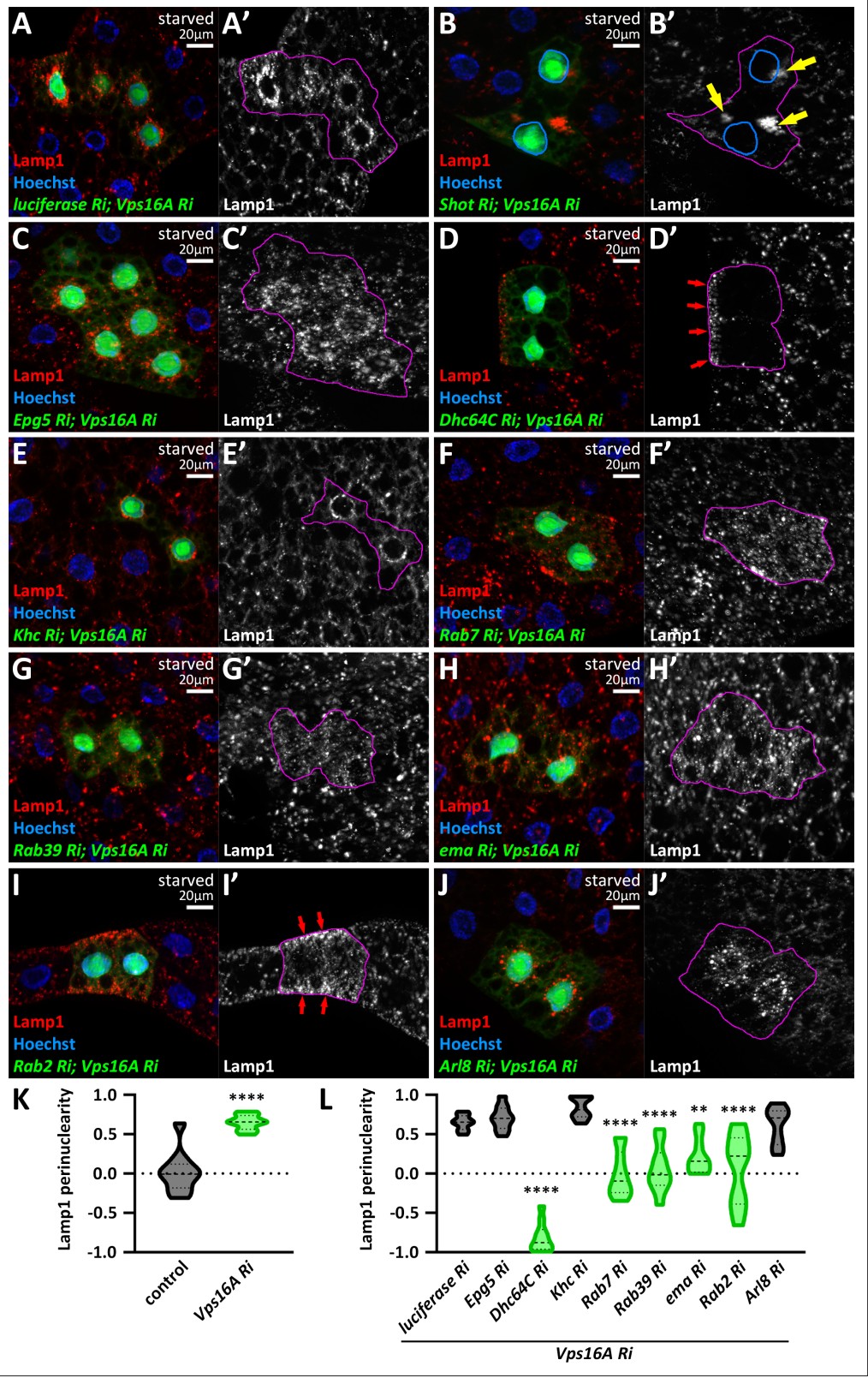

**Figure 7.** The positioning of pre-fusion, immature autolysosomes is very similar to autophagosomes in *vps16a* RNAi cells. (**A–J**) Lamp1-positive lysosomes accumulate around the nuclei in cells co-expressing a control (*luciferase*) RNAi (**A**). This positioning remains unaffected by the co-expression of *khc* (**E**) or *arl8* RNAi (**J**). Depletion of Shot (**B**) or Dhc64C (**C**) in *vps16a* RNAi cells redistributes Lamp1-positive lysosomes from the perinuclear

*Figure 7 continued on next page*

*Figure 7 continued*

cytoplasm to an ectopic microtubule organizing center (MTOC) (yellow arrows) or to the periphery, respectively. Lamp1-positive lysosomes are scattered throughout the cytosol in *vps16a* RNAi cells upon the co-expression of *rab7* (**F**), *rab39* (**G**), and *ema* (**H**) RNAi-s. In contrast, Lamp1-positive lysosomes retain their perinuclear distribution in *epg5, vps16a* double RNAi cells (**D**). Similar to Rab7 or Rab39, the expression of *rab2* RNAi in Vps16A KD cells results in the scattering of Lamp1-positive lysosomes, with a trend observed that lysosomes tend to accumulate near the periphery (red arrows) (**I**). The boundaries of RNAi-expressing cells are highlighted in magenta in the grayscale panels. The outlines of nuclei are drawn in blue in **B'**. (**K, L**) Quantification of data shown in **A, C-J**; n=10 cells.

lysosomal markers (*Byrne et al., 2016*; *Hegedűs et al., 2016*; *Wang et al., 2016*; *Figure 10F–H*, *Figure 10—figure supplement 1E–G*). Although loss of minus-end directed transport significantly impaired autophagosome-lysosome fusion, the degree of inhibition did not reach the level observed upon Vps16A knockdown (*Figure 10—figure supplement 1H–K*). This indicates that impaired motility alone does not directly block fusion, but rather reduces its probability. Taken together, our results suggest that the role of minus-end-directed transport in fat cells is to bring pre-fusion organelles into proximity to increase the likelihood of their fusion.

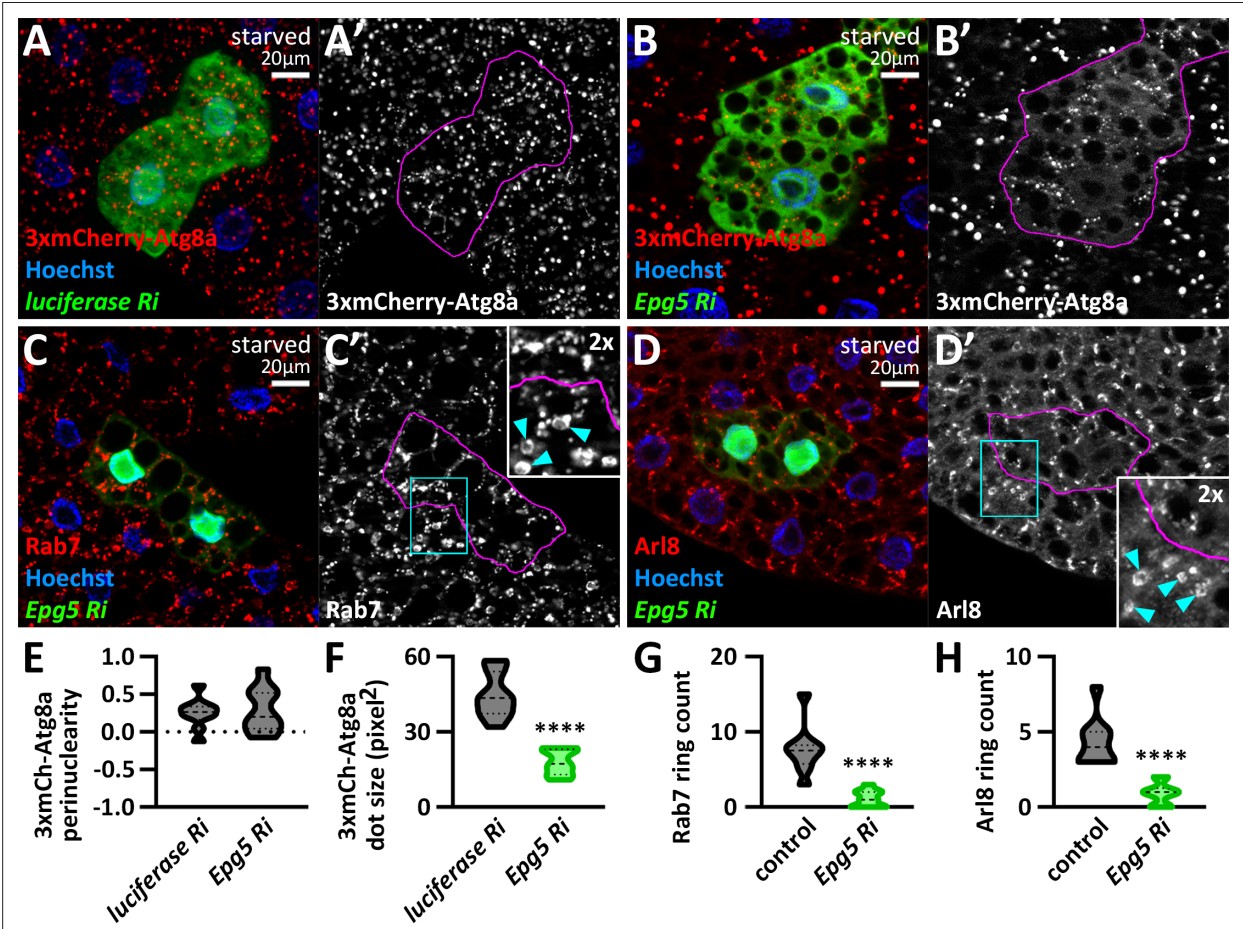

**Figure 8.** Epg5 regulates autolysosome maturation. (**A, B**) Epg5 knockdown results in a significant reduction in the size of 3xmCherry-Atg8a-positive autolysosomes (**B**) compared to control RNAi (*luciferase* RNAi) expressing cells (**A**). (**C, D**) *epg5* RNAi cells lack large Rab7 (**C**) and Arl8 (**D**) positive autolysosomes, which are present in surrounding control cells (cyan arrowheads in insets point to Rab7 and Arl8-positive autolysosomes in control cells). The boundaries of RNAi cells are highlighted in magenta in the grayscale panels. (**E–H**) Quantification of data shown in **A-D**; n=10 cells.

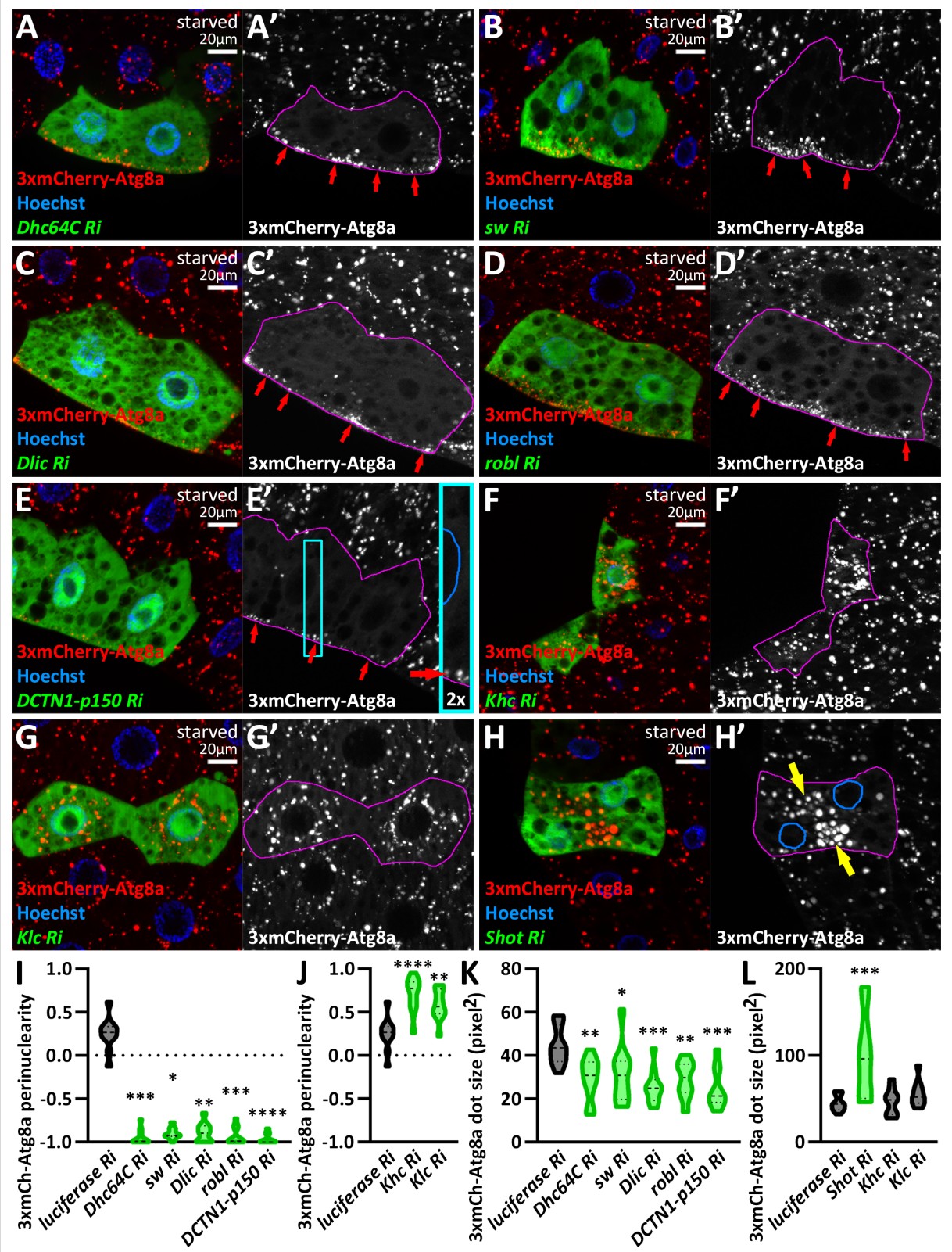

**Figure 9.** Minus-end-directed transport is required for autolysosome maturation. (**A–E**) Autolysosome size is significantly reduced upon the loss of dynein (**A–D**) or dynactin (**E**) function. Red arrows point to autolysosomes at the cell periphery in (**A'-E'**) and in the inset of E. (**F, G**) Kinesin knockdowns do not significantly influence autolysosome size. (**H**) Autolysosome size increases at ectopic foci (yellow arrows) in *shot* RNAi cells. The boundaries

*Figure 9 continued on next page*

Figure 9 continued

of RNAi-expressing cells are highlighted in magenta in the grayscale panels. The outlines of nuclei are drawn in blue in the inset of E' and in H'. (**I-L**) Quantification of data shown in **A-H**; n=10 cells.

The online version of this article includes the following figure supplement(s) for figure 9:

**Figure supplement 1.** Additional data on the effects of knocking down autophagosome transport machinery on autolysosomes.

## Discussion

Microtubular transport of different organelles within the lysosomal system is essential for their proper function. Among the small GTPases that regulate late endosome/lysosome positioning, Rab7 (*Fujiwara et al., 2016*; *Jordens et al., 2001*; *Ma et al., 2018*; *van der Kant et al., 2013*) and Arl8 (*Bagshaw et al., 2006*; *Boda et al., 2019*; *Hofmann and Munro, 2006*; *Marwaha et al., 2017*; *Rosa-Ferreira and Munro, 2011*; *Rosa-Ferreira et al., 2018*) are well-studied. However, the positioning of autophagosomes is less understood. Autophagosome transport has been primarily studied in highly polarized neurons under basal conditions. In neuronal cells, dynein-mediated transport, regulated by several suggested adaptor molecules, has been described. However, most reporters and reagents used for studying autophagosome transport cannot differentiate between pre-fusion and post-fusion autophagic vesicles. Therefore, the regulation of non-fused autophagosome motility remains unclear.

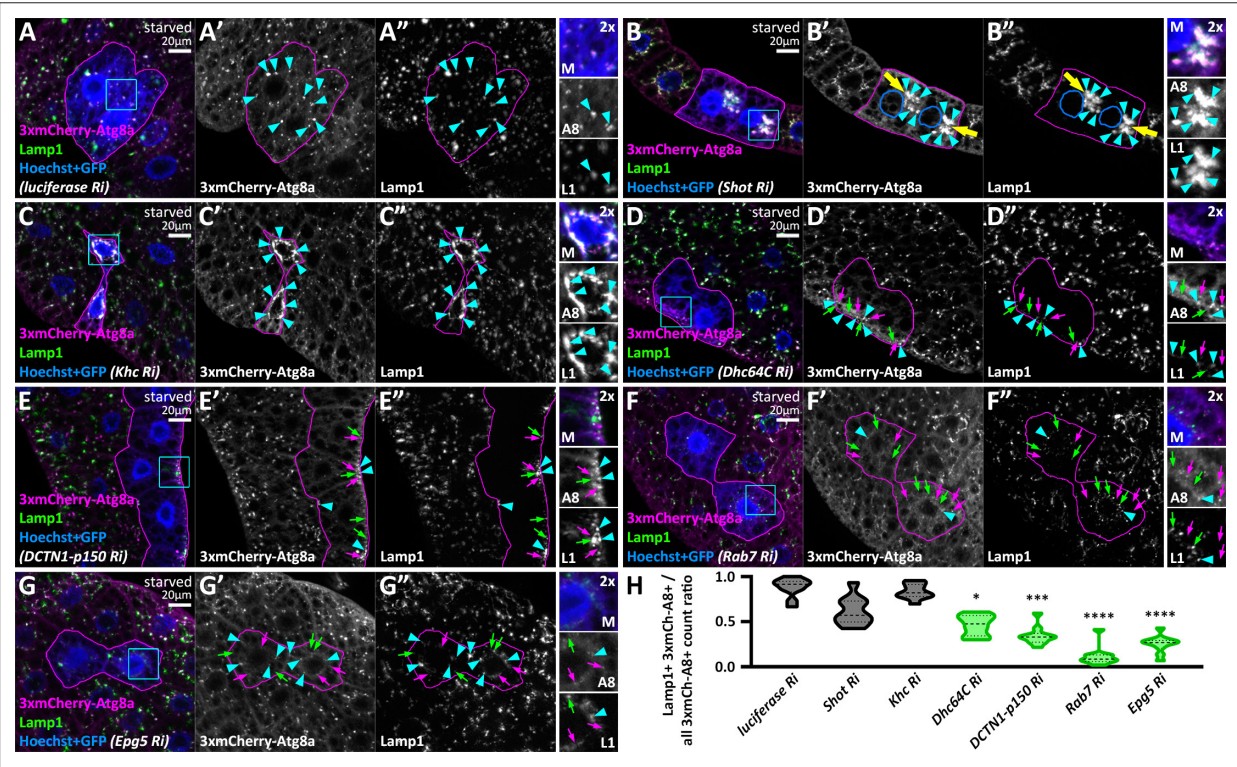

**Figure 10.** Loss of the autophagosome positioning machinery decreases autophagosome-lysosome fusion. (**A–G**) In starved control RNAi (*luciferase*) expressing cells, mCherry-Atg8a overlaps with endogenous Lamp1 (**A**), indicating normal autophagosome-lysosome fusion and autolysosome formation. Autolysosomes still form in *shot* (**B**) or *khc* RNAi (**C**) cells, as mCherry-Atg8a colocalizes with endogenous Lamp1 similar to controls, but these are found in ectopic foci (yellow arrows) in *shot* RNAi cells. The outlines of nuclei are drawn in blue in B' and B''. Conversely, RNAi-s targeting factors responsible for minus-end directed autophagosome transport (**D–G**) decrease this overlap, suggesting less effective autophagosome-lysosome fusion. The GFP signal of RNAi-expressing cells is false-colored blue in composite images. The boundaries of RNAi-expressing cells are highlighted in magenta. The boxed areas in the main panels, marked by cyan, are enlarged in the insets (M, merged image; A8, 3xmCherry-Atg8a; L1, Lamp1). Cyan arrowheads point to mCherry-Atg8a/Lamp1 double-positive structures, while magenta and green arrows indicate mCherry-Atg8a or Lamp1 single-positive dots, respectively. (**H**) Quantification of data shown in A-G; n=10 cells.

The online version of this article includes the following figure supplement(s) for figure 10:

**Figure supplement 1.** Additional data on the effects of knocking down autophagosome transport machinery on autophagosome-lysosome fusion.

We propose that an autophagosome should not be considered as such once its lumen has started to acidify or undergo fusion, transitioning to a maturing autolysosome.

To address this issue, we established a genetic system generating mosaic cells where Vps16A, a central HOPS subunit, is silenced. Without Vps16A, cells are unable to fuse autophagosomes with lysosomes or late endosomes (*Takáts et al., 2014*), thus preventing the formation of autolysosomes. This system allowed us to study autophagosome positioning without misidentifying autolysosomes as autophagosomes. Given that fat cells have perinuclear non-centrosomal MTOCs, we hypothesized that non-fused autophagosomes move in a minus-end direction, likely driven by dyneins. This hypothesis was supported by our observation that relocating the ncMTOC using *shot* RNAi also relocated autophagosomes to this region.

Loss of dyneins and dynactins not only blocked perinuclear autophagosome positioning but also redistributed them to the cell periphery, suggesting that kinesin-regulated motility becomes available for starvation-induced autophagosomes in these conditions. Importantly, we showed that not all dyneins or dynactins are required for this transport. A dedicated cytosolic dynein complex, composed of Dhc64 (a heavy chain subunit), sw (an intermediate chain subunit), Dlic (a light intermediate chain subunit), and robl (a light chain subunit), activated by a dynactin complex, is required for autophagosome transport. This is consistent with observations that autophagosomes can move bidirectionally in neurons and that purified autophagosome fractions contain both dyneins and kinesins (*Maday et al., 2012*). Our results also reinforce findings that microtubule inhibitors block centrosome-directed autophagosome transport in mammalian cells (*Fass et al., 2006*).

Taking these observations into consideration, we investigated the relationship between dyneins and kinesins in autophagosome transport. Overexpression of kinesin motors blocked minus-end transport, while autophagosomes in kinesin and dynactin double knockdown cells appeared immobile. Moreover, expressing a recombinant minus-end kinesin (*Clark et al., 1997*) partially rescued the peripheral relocation of autophagosomes upon dynactin silencing, indicating a competitive relationship between minus- and plus-end motors in autophagosome positioning. The possibility of plus-end transport as a secondary mechanism raises questions about its physiological role. Besides enabling bidirectional movement (*Jahreiss et al., 2008*; *Maday et al., 2012*), it is possible that autophagosomes transport kinesins to autolysosomes, similar to endosomes, which are suggested to transport dyneins to autophagic vacuoles (*Cheng et al., 2015*). Therefore, if kinesins are present but are downregulated on autophagosomes, the absence of dyneins could potentially release them from inhibition. Autophagic vesicles are suggested to move towards the plus end (*Mauvezin et al., 2016*; *Pankiv et al., 2010*), and loss of various kinesins leads to autophagosome accumulation in the cell center in mammalian cells (*Cardoso et al., 2009*; *Korolchuk et al., 2011*). The plus-end transport of autophagic vesicles by the Klp98A kinesin in *Drosophila* promotes autophagosome-lysosome fusion and degradation (*Mauvezin et al., 2016*). However, our tests indicated that Rab14 and its interactor Klp98A likely transport autolysosomes, not autophagosomes, as evidenced by the non-autophagosomal localization of YFP-Rab14 and that their RNAi-s had no effect on autophagosome positioning in starved, HOPS-depleted cells. Since neither kinesin RNAi altered the perinuclear accumulation of autophagosomes in Vps16A-depleted cells, we propose that the default direction is towards the MTOC at the minus-ends of microtubules.

Among Rab small GTPases, we identified Rab7 and Rab39 as regulators of autophagosome positioning. Rab7, crucial for autophagosome-lysosome fusion, appears on autophagosomes (*Hegedűs et al., 2016*). Its knockdown resulted in autophagosomes remaining randomly positioned in the cytosol, highlighting its importance in bidirectional motility. Consistent with our findings, Rab7 regulates both minus- and plus-end directed motility of lysosomes or endosomes, involving several adaptors such as Plekhm1 and FYCO1 (*Fujiwara et al., 2016*; *Jordens et al., 2001*; *Ma et al., 2018*; *Pankiv et al., 2010*; *Tabata et al., 2010*; *van der Kant et al., 2013*). However, only one Rab7 interactor, the autophagy adaptor Epg5 (*Gillingham et al., 2014*; *Wang et al., 2016*), produced a similar phenotype to Rab7 in our tests. Additionally, we demonstrated that Epg5 coprecipitates with Dhc64C, and localizes to Rab7 and Atg8a-positive vesicles in cultured fruit fly cells. As Epg5 loss did not significantly impact the endolysosomal system, it appears to function in the autophagic pathway. Epg5 interacts with Rab7 and LC3 to mediate autophagosome-lysosome fusion in fly, worm, and mammalian cells (*Hori et al., 2017*; *Wang et al., 2016*), and its mutations are linked to Vici syndrome, a severe neurodegenerative disorder in humans (*Balasubramaniam et al., 2018*;

*Byrne et al., 2016*; *Meneghetti et al., 2019*). We thus identify a potential new role for Epg5 in autophagosome positioning.

Silencing Rab39 and its interactor ema (*Gillingham et al., 2014*) produced a phenotype similar to Rab7 or Epg5 loss. Rab39 regulates lysosomal function and interacts with HOPS in mammalian cells (*Lakatos et al., 2021*; *Zhang et al., 2023*). Ema promotes autophagosome biogenesis (*Kim et al., 2012*) and endosomal maturation through HOPS interaction (*Kim et al., 2010*). Our results suggest that Rab7-Epg5 and Rab39-ema interactions are both necessary for bidirectional autophagosome motility. Further studies are required to clarify their exact roles and interrelations. YFP-tagged Rab7 and Rab39 both colocalized with autophagosomes, supporting their role in motility. Rab2, Rab7, and Arl8 are known lysosomal fusion factors (*Boda et al., 2019*; *Hegedűs et al., 2016*; *Lőrincz et al., 2017b*), but unlike Rab7, neither Rab2 nor Arl8 knockdown affected autophagosome positioning. Since YFP-Rab2 was also found to colocalize with autophagosomes, our results suggest that its sole role on autophagosomes is to regulate maturation. Although our RNAi screen identified a molecular machinery potentially sufficient for autophagosome transport, other small GTPases, adaptors, or motor subunits may also influence this process. Because validating the proper loss-of-function effect for every RNAi line used in the screen was impractical, we cannot exclude the possibility that some negative results were due to inefficient silencing. It is also important to note that *Drosophila* cells exhibit a wide variety of microtubule-organizing centers (MTOCs) during development, including both centrosomal and non-centrosomal types (*Tillery et al., 2018*). A key future direction will be to examine autophagosome motility in additional cell types to determine whether the same positioning machinery operates universally, or whether cell-type–specific mechanisms exist.

Given the small Arl8-positive lysosomes in the perinuclear region upon Vps16A silencing, we hypothesized that pre-fusion lysosomes might exhibit similar motility to autophagosomes. This was largely confirmed, as minus-end transport of immature lysosomes depended on dyneins, Rab7, Rab39, and ema. However, Epg5 did not influence lysosome positioning, indicating that its function is likely exerted at the surface of autophagosomes. Conversely, Rab2 was identified as a regulator of minus-end-directed lysosome transport. These findings suggest similar but distinct regulatory mechanisms for pre-fusion organelle transport, possibly to enhance fusion efficiency by converging them towards the cell center. Rab2's interaction with motor adaptors such as Bicaudal D (*Gillingham et al., 2014*) likely regulates lysosome motility.

Moreover, mature autolysosomes, but not pre-fusion ones, redistributed to the cell periphery upon Rab39 silencing, similar to dynein-depleted cells, suggesting that Rab39 exclusively regulates minus-end movement at the post-fusion level. Considering that plus-end directed lysosome motility is regulated by Arl8 (*Bagshaw et al., 2006*; *Boda et al., 2019*; *Hofmann and Munro, 2006*; *Marwaha et al., 2017*; *Rosa-Ferreira and Munro, 2011*; *Rosa-Ferreira et al., 2018*), which is dispensable for autophagosome motility, it is plausible that Rab39 promotes bidirectional transport of pre-fusion organelles, while its role post-fusion is restricted to minus-end directed motility.

An important question is why pre-fusion autophagosomes and lysosomes travel to the same destination. It is feasible to think this is because they need to fuse with each other. We found that loss of minus-end directed motility reduced autolysosome maturation; in dynein- or dynactin-depleted cells, only smaller autolysosomes were produced, and autophagosome-lysosome fusion decreased. This aligns with previous observations that dynein-regulated autophagosomal motility is indispensable for efficient lysosomal fusion (*Jahreiss et al., 2008*; *Kimura et al., 2008*). Conversely, inhibiting kinesins did not impede autophagosome-lysosome fusion, and concentrating autophagosomes and lysosomes to a smaller ectopic ncMTOC via *shot* RNAi resulted in enlarged lysosomes, allowing autophagosome-lysosome fusion to proceed.

Therefore, we propose a model in which pre-fusion organelles travel towards the MTOC in a cytosolic dynein-dependent manner regulated by small GTPases and their adaptors (Rab7-Epg5 and Rab39-ema on autophagosomes; Rab7, Rab39-ema, and Rab2 on lysosomes) to enhance fusion probability. After fusion, autolysosomes can move to the periphery in an Arl8-dependent manner and back, regulated by Rab39 (*Figure 11*).

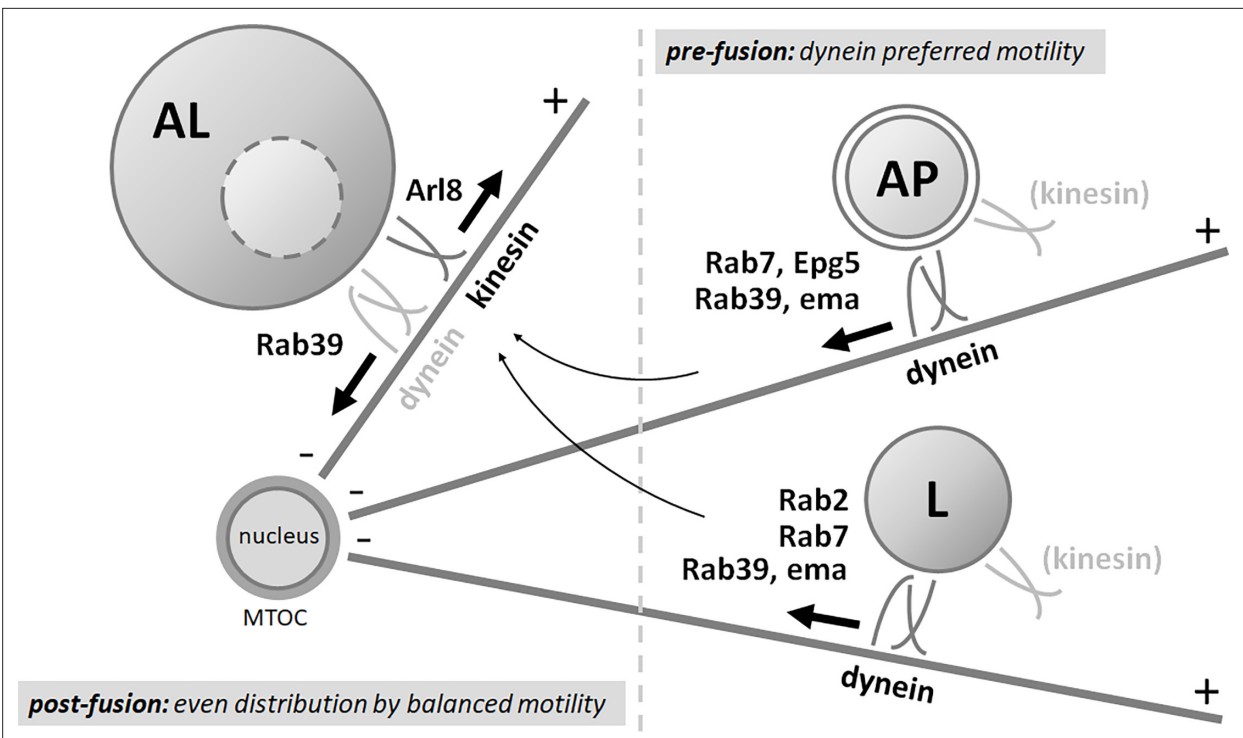

**Figure 11.** Model of the transport of autophagosomes and lysosomes in starved fat cells. Before fusion, autophagosomes and lysosomes are transported towards the perinuclear non-centrosomal microtubule organizing center (ncMTOC) by a cytosolic dynein complex in starved fat cells to ensure proper fusion and effective degradation. This process requires Rab7, Rab39, and their interactors Epg5 and ema on autophagosomes, and Rab2, Rab7, Rab39, and the Rab39 interactor ema, but not Epg5, on lysosomes. After fusion, Arl8 mediates the plus-end transport of autolysosomes, while Rab39 promotes dynein-regulated minus-end directed transport. Thus, the motility of pre-fusion and post-fusion organelles is differently regulated: pre-fusion organelles generally move towards the MTOC, while post-fusion organelles exhibit bidirectional motility. This spatial regulation ensures proper fusion rates and degradation efficiency.

## Materials and methods
### Fly work and RNAi-based screen

We raised the fly stocks and crosses in glass vials, on standard food at 25 °C. Early third instar larvae were starved for 3 hr in 20% sucrose solution. Next, fat bodies were dissected in cold PBS, mounted in an 8:2 mixture of glycerol and PBS completed with Hoechst 33342 as nuclear dye (5 µg/ml) (Thermo), then imaged immediately.

For the RNAi-based genetic screen, we established the *hs-Flp; Vps16A RNAi, UAS-DCR2; act <CD2<Gal4, UAS-GFPnls, r4-mCherry-Atg8a* stock, in order to generate *vps16a* RNAi-expressing fat cells. This was crossed with RNAi or overexpression lines of interest.

All the screened *Drosophila* lines, as well as their sources, identifiers, and phenotypes are listed in *Supplementary file 2*. Representative images of the phenotypes of screened lines are shown in *Supplementary file 1*. The proper genotypes and the fly stocks from the screen that were used in the Figure panels are summarized in *Supplementary file 3*.

For further experiments, we used the following mosaic cell-generating stocks, with or without Vps16A RNAi:

- *hs-Flp; Vps16A RNAi, UAS-DCR2; act <CD2<Gal4, UAS-GFPnls,*
- *hs-Flp; UAS-DCR2; act <CD2<Gal4, UAS-GFPnls,*
- *hs-Flp; 3xmCherry-Atg8a, UAS-2xEGFP; act <CD2<Gal4, UAS-DCR2,*
- *hs-Flp; 3xmCherry-Atg8a, UAS-GFP-Lamp1; act <CD2<Gal4, UAS-DCR2.*

All of these stocks were described before (*Boda et al., 2019*; *Lőrincz et al., 2017b*; *Takáts et al., 2013*). The *prospector-Gal4* driver (80572; FlyBase ID: FBst0080572) and the *UAS-GFP-myc-2xFYVE*

reporter (42712; FlyBase ID: FBst0042712) used for the garland cell experiments were obtained from Bloomington *Drosophila* Stock Center.

## Immunohistochemistry

For immunohistochemistry experiments, we dissected and fixed the samples in 4% paraformaldehyde (in PBS) for 45 min, washed them in PBS for 2×15 min, permeabilized in PBTX (0.1% Triton X-100 in PBS) for 20 min, and blocked in 5% fetal bovine serum (in PBTX) for 30 min. The samples were then incubated with the primary antibodies (diluted in the blocking solution) overnight at 4 °C, followed by washing in PBTX containing 4% NaCl for 15 min, washing in PBTX for 2×15 min, blocking in 5% fetal bovine serum (in PBTX) for 30 min and incubating with the secondary antibodies for 3 hr. The samples were then washed in PBTX containing 4% NaCl and 5 µg/ml Hoechst 33342 for 15 min, in PBTX for 2×15 min, and in PBS for 2×15 min. All steps, except the incubation with primary antibodies, were performed at room temperature.

In case of garland immunohistochemistry and Lamp1 immunostainings of fat bodies for the 3xmCherry-Atg8a colocalization experiment, samples were dissected in a buffer containing 80 mM PIPES, 5 mM EGTA, and 1 mM $MgCl_2$ (pH was adjusted to 6.8 with NaOH) and fixed in this solution containing also 3.7% formaldehyde, 0.25% glutaraldehyde, and 0.2% Triton X-100, for 45 min. Following the fixation, the samples were incubated with 2 mg/ml sodium borohydride (in PBS) for 2.5 min, washed in PBS for 2×15 min (once for 10 min in case of garland cells) and permeabilized in PBTX containing ammonium chloride and glycine (both 50 mM) for 20 min. The remaining steps were the same as described above.

For immunostaining S2*R*+ hemocytes (*Drosophila*), cells were fixed in 4% paraformaldehyde for 20 min, washed in PBS for 15 min, permeabilized in PBTX for 10 min, and blocked in 5% fetal bovine serum (in PBTX) for 30 min. The samples were then incubated with the primary antibodies overnight at 4 °C, followed by washing in PBTX containing 4% NaCl for 15 min and in PBTX for 2×10 min, and incubation with the secondary antibodies (solved in the blocking solution) for 3 hr. The cells were then washed in PBTX containing 4% NaCl and 5 µg/ml Hoechst 33342 for 15 min, in PBTX for 10 min, and in PBS for 15 min. In case of Atg8a immunolabeling, cells were starved in a solution containing 10 mM D(+) glucose, 0.5 mM $MgCl_2$, 4.5 mM KCl, 121 mM NaCl, 0.7 mM $Na_2HPO_4$, 1.5 mM $NaH_2PO_4$, and 15 mM $NaHCO_3$ (pH 7.4) (*Aguilera-Gomez et al., 2017*).

The following primary antibodies were used: rat anti-Atg8a (1:800 or in case of S2*R*+ cells, 1:300; *Takáts et al., 2013*); rabbit anti-β-galactosidase (1:100; Merck); goat anti-Gmap (1:1000; Developmental Studies Hybridoma Bank [DSHB]); mouse anti-Rab7 (1:10; DSHB; *Riedel et al., 2016*); rabbit anti-Arl8 (1:300; DSHB); rabbit anti-Lamp1 (1:1000; *Chaudhry et al., 2022*), rat anti-mCherry (1:300; *Takáts et al., 2014*); guinea pig anti-mCherry (1:500); rabbit anti-HA (1:200; Merck); rabbit anti-HA (1:200; Proteintech) and chicken anti-GFP (1:1500; Invitrogen).

We used the following secondary antibodies: Alexa Fluor 568 goat anti-rat (1:1000); Alexa Fluor 647 donkey anti-rabbit (1:600); Alexa Fluor 568 donkey anti-goat (1:1000); Alexa Fluor 568 donkey anti-mouse (1:1000); Alexa Fluor 647 donkey anti-mouse (1:600); Alexa Fluor 568 donkey anti-rabbit (1:1000); Alexa Fluor 488 goat anti-chicken (1:1000); Alexa Fluor 488 donkey anti-rat (1:1000); Alexa Fluor 488 goat anti-rabbit (1:1000) (all Invitrogen) and DyLight 550 goat anti-guinea pig (1:600; Thermo Fisher).

## Electron microscopy

For correlative ultrastructural analysis, fat bodies were dissected in a fixative containing 3.2% paraformaldehyde, 1% glutaraldehyde, 1% sucrose, and 0.003 M $CaCl_2$ (in 0.1 N sodium cacodylate buffer, pH 7.4) on poly-L-lysine-coated glass slides and fluorescent images were taken to help recognize GFP and RNAi-expressing cells. Then the samples were fixed in the same solution overnight at 4 °C, then were post-fixed in 0.5% osmium tetroxide for 1 hr, followed by half-saturated aqueous uranyl acetate for 30 min and dehydrated in a graded series of ethanol, followed by embedding into Durcupan ACM (Sigma-Aldrich) on the glass slides. RNAi cells in the embedded samples were identified in semi-thick sections stained with toluidine blue, then ultra-thin sections of 70 nm were cut, then stained with Reynold's lead citrate (8 min, RT).

Preparation of samples for ultrastructural analysis of garland nephrocytes was performed as described before (*Lőrincz et al., 2016*).

Images were taken by a JEOL JEM-1011 transmission electron microscope operating at 80 kV, equipped with a Morada camera (Olympus) and iTEM software (Olympus).

## Molecular cloning and biochemistry

### Cloning

To generate *genEpg5-9xHA*, genomic DNA from *w1118 Drosophila* strain was isolated and used as a template. The genomic region containing *Drosophila CG14299* was amplified using primers 5'-CCAA GCTTGCATGCGGCCGCATTTTCTGTGCGCGACTGTTG-3' and 5'-TAAAAGATGCGGCCGGTACC GCCTCCACCCGTGGCCATTAACTTGAATTC-3' and cloned into *pGen-9xHA* (*Lőrincz et al., 2016*) as a NotI-Acc65I fragment by using the Gibson Assembly kit (New England BioLabs, Ipswich, MA).

To obtain N-terminally 3xFLAG-tagged Rab7, the coding region of Rab7 was amplified from *Drosophila* cDNA (GH03685 (DGRC Stock 7144; https://dgrc.bio.indiana.edu//stock/7144; RRID:DGRC_7144)) using primers 5'-ACAAGGCGGCCGCAGGTATGTCCGGACGTAAGAAATCC-3' and 5'-TCTAGAGGTACCTTAGCACTGACAGTTGTCAGGA-3' and cloned into NotI-Acc65I sites of *pUAST-3xFLAG* vector (*Takáts et al., 2014*).

### S2R+ maintenance and transfection

The S2*R+ Drosophila* cell line (*Drosophila* Genomics Resource Center; Stock 150; RRID:CVCL_Z831) was maintained in Insect XPress medium (Lonza) containing 10% FBS (EuroClone) and 1% Penicillin-Streptomycin (Lonza) at 26 °C. The cell line was not tested for mycoplasma contamination. Cells were transfected with *genEpg5-9xHA* plasmid using the calcium phosphate method. DNA was diluted in 240 mM $CaCl_2$, mixed with 2 x HEPES-buffered saline (50 mM HEPES, 1.5 mM $Na_2HPO_4$, 280 mM NaCl, pH 7.1), incubated at 25 °C for 30 min, and added to the cells. Co-transfection with *genEpg5-9xHA* and *genLamp1-3xmCherry* (*Hegedűs et al., 2016*) was performed using jetOPTIMUS DNA Transfection Reagent (Polyplus). 24 hr after transfection, cells were used for immunohistochemistry or immunoprecipitation. In experiments, when cells were transfected with *pUAST-3xFLAG-Rab7* and *pGen-Epg5-9xHA* constructs, metallothionein-Gal4 plasmid was also applied. Protein expression was induced 24 hr after transfection with 500 µM $CuSO_4$ for overnight incubation.

### Immunoprecipitation

Cells were transfected with appropriate plasmid constructs and were collected 24 hr after transfection. They were washed with PBS and lysed on ice in lysis buffer (0.5% Triton X-100, 150 mM NaCl, 5 mM EDTA, and 50 mM Tris-HCl, pH 7.5, complete protease inhibitor cocktail (Roche)) for 20 min. Cell lysates were cleared by centrifugation for 10 min at 20.000 g, 4 °C, followed by the addition of mouse anti-HA or anti-FLAG agarose (Sigma-Aldrich) to the supernatant. After incubation at 4 °C for 2 hr, beads were collected by centrifugation at 5.000 g for 2 min at 4 °C, followed by extensive washes in wash buffer (lysis buffer without protease inhibitors) and finally boiling in Laemmli sample buffer. Samples were analyzed by Western blot using rat anti-HA (1:1000; Roche), mouse anti-FLAG (M2; 1:2000; Sigma-Aldrich), and mouse anti-Dhc (2C11-2; 1:12.5; DSHB) antibodies. It is experimentally demonstrated that the Dhc antibody recognizes a polypeptide at around 260 kDa (*Baker et al., 2021*).

## Imaging, quantification, and statistics

We obtained the fluorescent images with an AxioImager M2 microscope (Zeiss), equipped with an ApoTome2 grid confocal unit (Zeiss) and with an Orca Flash 4.0 LT sCMOS camera (Hamamatsu), using Plan-Apochromat 40×/0.95 NA Air and Plan-Apochromat 63×/1.40 NA Oil objectives (Zeiss), and Zeiss Efficient Navigation 2 software. Images from eleven consecutive focal planes (section thickness: 0.35 µm in case of the 40× objective, and 0.25 µm in case of the 63×objective) were merged into one image. In case of S2*R+* cells, fluorescent images were obtained with an Olympus IX83 inverted fluorescent microscope, equipped with an Orca FusionBT CMOS camera (Hamamatsu), using a Universal Plan Extended Apochromat 60×/1.42 NA objective (Olympus), and cellSens Dimension 4.1 software (Olympus). Images were taken with full optical sectioning of the cells; the focal planes were merged into one image and deconvolution was applied. Figures were produced in Photoshop CS5 Extended (Adobe).

Fluorescent structures were quantified either using ImageJ software (National Institutes of Health) or, in case of some types of experiment, manually. The signal threshold of the fluorescent channel of interest was set by the same person during quantifying one type of experiment with ImageJ. The fat cells, S2R+ cells, and garland nephrocytes were randomly selected for quantification.

Structure distributions were quantified using ImageJ. In all cases, only cells with their nuclei in the focal plane were selected to ensure that both perinuclear and peripheral regions were included in quantifications. To quantify structure distribution, we divided the fat cells into a perinuclear and a peripheral domain that were measured to be equal and calculated the area of the fluorescent signal in both domains. Then the difference in signal areas of the perinuclear and peripheral domains was divided by the signal area of the entire cell, thus obtaining a ratio that represents the distribution of structures (1 – perfectly perinuclear, 0 – evenly dispersed, –1 – perfectly peripheral). In case of distribution of non-colocalizing 3xmCherry-Atg8a or GFP-Lamp1-positive structures, overlapping dots were removed from each channel using Photoshop CS5 Extended.

The counts of cells with ectopic ncMTOC (*shot* RNAi lines) and the Rab7/Arl8 ring counts (*epg5* RNAi) were determined manually by the same person. For quantifying Gmap or mCherry-Atg8a signal intensities, mean gray values of neighboring control and RNAi cells were calculated by ImageJ. Colocalizations were quantified manually by the same person in case of fat body cells. Dot plots and Pearson's coefficients were calculated by ImageJ for evaluating colocalization in S2R+ cells.

To evaluate data from garland nephrocyte experiments, we used ImageJ to quantify fluorescent structures from unmodified single focal planes. To quantify the size of the Rab7 and FYVE-GFP-positive endosomes, we measured the area of individual vesicles in the given focal plane of the cells. After setting the threshold for the fluorescent signal, we used the Watershed function of ImageJ coupled with manual segmentation when it was necessary to properly separate endosomes. For each genotype, we used 5 animals and measured the size of endosomes from a total of 10 cells. To quantify Lamp1 antibody staining signal in nephrocytes, we measured the area fraction of the cells covered by the given fluorescent signal at proper and uniform threshold settings. For each genotype, we used 5 late L3 stage animals and measured 15 cells.

Data were statistically evaluated using Prism 9.4.1 (GraphPad). The distribution of the datasets was determined using the D'Agostino & Pearson normality test. Parametric, unpaired, two-tailed t-test or one-way ANOVA (with Dunnett multiple comparisons test) was used to compare two or more samples, respectively, all showing normal distribution. When comparing two or more samples that contained at least one variable showing non-Gaussian distribution, we used non-parametric Mann-Whitney test or Kruskal-Wallis test (with Dunn's multiple comparisons test), respectively. We showed the data as violin plots in the figures and represented p-values as asterisks (<0.0001 ****; 0.0001–0.001 ***; 0.001–0.01 **; 0.01–0.05 *; 0.05<non-significant). Samples that are significantly different from the control are marked by green on the violin plots. All experiments were repeated on a different day, with similar results.

## Acknowledgements

We thank Sarolta Pálfia for technical assistance, and colleagues and stock centers mentioned in the Materials and methods section for supporting our work by providing fly stocks and reagents. This work has been implemented with the support provided by the Ministry of Culture and Innovation of Hungary from the National Research, Development and Innovation Fund (PD142943 to AB; FK138851 to PL; KKP129797 to GJ; New National Excellence Program: ÚNKP-23–3-I-ELTE-724 to DH, ÚNKP-23–3-I-ELTE-706 to MM; Doctoral Excellence Program: DKÖP-2023-ELTE-13 to DH), the Hungarian Academy of Sciences (LP2022-13/2022 to PL, LP2023-6 to GJ), and the Eötvös Loránd University Excellence Fund (EKA 2022/045-P101-2 to PL). The funders had no role in designing experiments, data collection and analysis, decision to publish, or preparation of the manuscript.

# Additional information

## Funding

| Funder | Grant reference number | Author |
|---|---|---|
| National Research, Development and Innovation Office | PD142943 | Attila Boda |
| National Research, Development and Innovation Office | FK138851 | Péter Lőrincz |
| National Research, Development and Innovation Office | KKP129797 | Gábor Juhász |
| National Research, Development and Innovation Office | ÚNKP-23-3-I-ELTE-724 | Dávid Hargitai |
| National Research, Development and Innovation Office | ÚNKP-23-3-I-ELTE-706 | Márton Molnár |
| National Research, Development and Innovation Office | DKÖP-2023-ELTE-13 | Dávid Hargitai |
| Magyar Tudományos Akadémia | LP2022-13/2022 | Péter Lőrincz |
| Magyar Tudományos Akadémia | LP2023-6 | Gábor Juhász |
| Eötvös Loránd Tudományegyetem | EKA 2022/045-P101-2 | Péter Lőrincz |

The funders had no role in study design, data collection and interpretation, or the decision to submit the work for publication.

## Author contributions

Attila Boda, Funding acquisition, Investigation, Visualization, Methodology, Writing – original draft, Writing – review and editing, BA performed experiments, contributed funding, designed some experiments, prepared figures, and wrote the original and revised manuscripts with input from all authors; Villő Balázs, Investigation, VB performed experiments and conducted fly crossings; Anikó Nagy, Investigation, AN performed experiments and conducted fly crossings; Dávid Hargitai, Funding acquisition, Investigation, HD performed garland nephrocyte experiments and contributed funding; Mónika Lippai, Investigation, ML performed experiments and conducted fly crossings; Zsófia Simon-Vecsei, Investigation, Methodology, Z-SV designed and performed S2R+ cell and co-immunoprecipitation experiments; Márton Molnár, Funding acquisition, Investigation, MM designed and performed S2R+ cell and co-immunoprecipitation experiments and contributed funding; Fanni Fürstenhoffer, Investigation, FF performed experiments and conducted fly crossings; Gábor Juhász, Resources, Funding acquisition, Writing – review and editing, GJ provided funding, reagents, and equipment, and reviewed and edited the manuscript; Péter Lőrincz, Conceptualization, Supervision, Funding acquisition, Investigation, Writing – original draft, Project administration, Writing – review and editing, PL conceived the main ideas and goals of the project, provided funding, designed experiments, and managed the research. PL supervised the project, performed TEM experiments, and wrote the original and revised manuscripts with input from all authors

## Author ORCIDs

Attila Boda ⓘ https://orcid.org/0000-0003-1811-8595
Zsófia Simon-Vecsei ⓘ https://orcid.org/0000-0001-7909-4895
Gábor Juhász ⓘ https://orcid.org/0000-0001-8548-8874
Péter Lőrincz ⓘ https://orcid.org/0000-0001-7374-667X

Reviewer #1 (Public review): https://doi.org/10.7554/eLife.102663.3.sa1
Reviewer #2 (Public review): https://doi.org/10.7554/eLife.102663.3.sa2
Reviewer #3 (Public review): https://doi.org/10.7554/eLife.102663.3.sa3
Author response https://doi.org/10.7554/eLife.102663.3.sa4

## Additional files

### Supplementary files

Supplementary file 1. Representative images of the phenotypes from all screened lines. The boundaries of silenced or overexpressing cells are highlighted in magenta, while positive hits are marked with green frames and captions.

Supplementary file 2. Detailed information about the screened lines, including their sources, identifiers, and phenotypes. The autophagosome distribution phenotypes are presented graphically for enhanced visibility (see the legend included in the table).

Supplementary file 3. Genotypes of the larvae and cells, along with a list of stocks from the screen used for the experiments shown in the figure panels.

MDAR checklist

Source data 1. Detailed statistical information for the experiments included in the figures.

### Data availability

All data generated or analysed during this study are included in the manuscript and supporting files.

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

# Appendix 1

## Appendix 1—key resources table

All the screened *Drosophila* lines, as well as their sources, identifiers, and phenotypes are listed in *Supplementary file 2*.

| Reagent type (species) or resource | Designation | Source or reference | Identifiers | Additional information |
|---|---|---|---|---|
| Strain, strain background (*D. melanogaster*) | 'hs-Flp; UAS-DCR2; act <CD2<Gal4, UAS-GFPnls;' <br> 'hs-Flp; 3xmCherry-Atg8a, UAS-2xEGFP; act <CD2<Gal4, UAS-DCR2;' <br> 'hs-Flp; 3xmCherry-Atg8a, UAS-GFP-Lamp1; act <CD2<Gal4, UAS-DCR2' | *Boda et al., 2019* https://doi.org/10.1016/j.bbamcr.2018.12.011; *Lőrincz et al., 2017b* https://doi.org/10.1083/jcb.201611027; *Takáts et al., 2013* https://doi.org/10.1083/jcb.201211160 | | |
| Genetic reagent (*D. melanogaster*) | *Vps16A RNAi GD* | Vienna *Drosophila* Resource Center (VDRC) | VDRC:23769; FLYB:FBst0455191 | |
| Genetic reagent (*D. melanogaster*) | *prospero-Gal4* | Bloomington *Drosophila* Stock Center (BDSC) | BDSC:80572; FLYB:FBst0080572 | |
| Genetic reagent (*D. melanogaster*) | *UAS-GFP-myc-2xFYVE* | BDSC | BDSC:42712; FLYB:FBst0042712 | |
| Cell line (*D. melanogaster*) | S2R+ | *Drosophila* Genomics Resource Center (Stock 150) | RRID:CVCL_Z831 | |
| Transfected construct (*D. melanogaster*) | *genEpg5-9xHA* | this paper | | Transfected construct (*D. melanogaster*) |
| Transfected construct (*D. melanogaster*) | *genLamp1-3xmCherry* | *Hegedűs et al., 2016* https://doi.org/10.1091/mbc.E16-03-0205 | FLYB:FBtp0116217 | Transfected construct (*D. melanogaster*) |
| Antibody | anti-Atg8a (Rat monoclonal) | *Takáts et al., 2013* https://doi.org/10.1083/jcb.201211160 | | IF(1:800); IF in case of S2R+ cells(1:300) |
| Antibody | anti-β-galactosidase (Rabbit monoclonal) | ZooMAb (Sigma-Aldrich) | Cat# ZRB1700 | IF(1:100) |
| Antibody | anti-Gmap (Goat polyclonal) | Developmental Studies Hybridoma Bank (DSHB) | Cat# GMAP; RRID:AB_2618259 | IF(1:1000) |
| Antibody | anti-Rab7 (Mouse monoclonal) | DSHB; *Riedel et al., 2016* https://doi.org/10.1242/bio.018937 | Cat# Rab7; RRID:AB_2722471 | IF(1:10) |
| Antibody | anti-Arl8 (Rabbit polyclonal) | DSHB | Cat# Arl8; RRID:AB_2618258 | IF(1:300) |
| Antibody | anti-Lamp1 (Rabbit polyclonal) | *Chaudhry et al., 2022* https://doi.org/10.1080/15548627.2022.2038999 | | IF(1:1000); Andreas Jenny |
| Antibody | anti-mCherry (Rat polyclonal) | *Takáts et al., 2014* https://doi.org/10.1091/mbc.E13-08-0449 | | IF(1:300) |
| Antibody | anti-mCherry (Guinea pig polyclonal) | *Hegedűs et al., 2016* | | IF(1:500); Gábor Juhász |
| Antibody | anti-HA (Rabbit polyclonal) | Sigma-Aldrich | Cat# H6908 | IF(1:100) |
| Antibody | anti-HA (Rabbit polyclonal) | Proteintech | Cat# 51064–2-AP | IF(1:200) |
| Antibody | anti-HA (Rat monoclonal) | Roche | Cat# 3F10 | WB(1:1000) |
| Antibody | anti-GFP (Chicken polyclonal) | Invitrogen | Cat# A10262 | IF(1:1500) |
| Antibody | anti-FLAG M2 (Mouse monoclonal) | Sigma-Aldrich | Cat# F1804 | WB(1:2000) |
| Antibody | anti-Dhc (Mouse monoclonal) | DSHB | Cat# 2C11-2; RRID:AB_2091523 | WB(1:12.5) |
| Antibody | Alexa Fluor 568 anti-Rat (Goat polyclonal) | Invitrogen | Cat# A-11077 | IF(1:1000) |
| Antibody | Alexa Fluor 647 anti-Rabbit (Donkey polyclonal) | Invitrogen | Cat# A-31573 | IF(1:600) |

*Appendix 1 Continued on next page*

*Appendix 1 Continued*

| Reagent type (species) or resource | Designation | Source or reference | Identifiers | Additional information |
|---|---|---|---|---|
| Antibody | Alexa Fluor 568 anti-Goat (Donkey polyclonal) | Invitrogen | Cat# A-11057 | IF(1:1000) |
| Antibody | Alexa Fluor 568 anti-Mouse (Donkey polyclonal) | Invitrogen | Cat# A10037 | IF(1:1000) |
| Antibody | Alexa Fluor 647 anti-Mouse (Donkey polyclonal) | Invitrogen | Cat# A-31571 | IF(1:600) |
| Antibody | Alexa Fluor 568 anti-Rabbit (Donkey polyclonal) | Invitrogen | Cat# A10042 | IF(1:1000) |
| Antibody | Alexa Fluor 488 anti-Chicken (Goat polyclonal) | Invitrogen | Cat# A-11039 | IF(1:1000) |
| Antibody | Alexa Fluor 488 anti-Rat (Donkey polyclonal) | Invitrogen | Cat# A-21208 | IF(1:1000) |
| Antibody | Alexa Fluor 488 anti-Rabbit (Goat polyclonal) | Abcam | Cat# ab150077 | IF(1:1000) |
| Antibody | DyLight 550 anti-Guinea pig (Goat polyclonal) | Thermo Fisher | Cat# SA5-10095 | IF(1:600) |
| Antibody | anti-Rat-HRP (Goat polyclonal) | Sigma-Aldrich | Cat# A9037 | WB(1:4000) |
| Antibody | anti-Mouse-HRP (Rabbit polyclonal) | Sigma-Aldrich | Cat# A9044 | WB(1:10000) |
| Recombinant DNA reagent | *pGen-9xHA* | **Lőrincz et al., 2016** https://doi.org/10.7554/eLife.14226 | | plasmid |
| Recombinant DNA reagent | *pUAST-3xFLAG* | **Takáts et al., 2014** https://doi.org/10.1091/mbc.E13-08-0449 | | plasmid |
| Recombinant DNA reagent | *metallothionein-Gal4* | **Takáts et al., 2013** https://doi.org/10.1083/jcb.201211160 | | plasmid |
| Recombinant DNA reagent (*D. melanogaster*) | Rab7 cDNA clone | *Drosophila* Genomics Resource Center (DGRC) | DGRC Stock Number: 7144; RRID:DGRC_7144 | cDNA |
| Sequence-based reagent | genEpg5_F | this paper | PCR primers | CCAAGCTTGCATGCGGCCGCA TTTTCTGTGCGCGACTGTTG |
| Sequence-based reagent | genEpg5_R | this paper | PCR primers | TAAAAGATGCGGCCGGTACCGCCT CCACCCGTGGCCATTAACTTGAATTC |
| Sequence-based reagent | Rab7CDS_F | this paper | PCR primers | ACAAGGCGGCCGCAGGTAT GTCCGGACGTAAGAAATCC |
| Sequence-based reagent | Rab7CDS_R | this paper | PCR primers | TCTAGAGGTACCTTAGCA CTGACAGTTGTCAGGA |
| Commercial assay or kit | Gibson Assembly kit | New England BioLabs | Cat# E5510S | |
| Commercial assay or kit | Durcupan ACM | Sigma-Aldrich | Cat# 44610 | |
| Software, algorithm | Photoshop CS5 Extended 12.1x64 | Adobe | RRID:SCR_014199 | |
| Software, algorithm | Prism 9.4.1 | GraphPad | RRID:SCR_002798 | |
| Software, algorithm | Zeiss Efficient Navigation 2 | Zeiss | RRID:SCR_021725 | |
| Software, algorithm | cellSens Dimension 4.1 | Olympus | RRID:SCR_014551 | |
| Software, algorithm | ImageJ 1.50b Fiji | National Institutes of Health, USA | RRID:SCR_003070 | |
| Software, algorithm | iTEM | Olympus | | |
| Other | Hoechst 33342 | Thermo Fisher | Cat# 62249 | Nuclear dye; 5 µg/ml |
| Other | jetOPTIMUS DNA Transfection Reagent | Polyplus | Cat# 101000051 | See Molecular cloning and biochemistry, S2R+ maintenance and transfection subsection in the Materials and methods |

*Appendix 1 Continued on next page*

*Appendix 1 Continued*

| Reagent type (species) or resource | Designation | Source or reference | Identifiers | Additional information |
|---|---|---|---|---|
| Other | Insect-XPRESS Protein-free Insect Cell Medium with L-glutamine | Lonza | Cat# 12-730Q | See Molecular cloning and biochemistry, S2R+ maintenance and transfection subsection in the Materials and methods |
| Other | complete protease inhibitor cocktail | Roche | Cat# COEDTAF-RO | See Molecular cloning and biochemistry, Immunoprecipitation subsection in the Materials and methods |
| Other | anti-HA agarose (Mouse monoclonal) | Millipore | Cat# A2095 | See Molecular cloning and biochemistry, Immunoprecipitation subsection in the Materials and methods |
| Other | anti-FLAG agarose (Mouse monoclonal) | Millipore | Cat# A2220 | See Molecular cloning and biochemistry, Immunoprecipitation subsection in the Materials and methods |

