## [Editor Report · eLife Assessment]

This paper presents **valuable** findings on how autophagosomes are positioned along microtubules for their efficient fusion with lysosomes, providing significant insights into the mechanism. The evidence supporting the conclusions is **solid**, with high-quality fluorescence microscopy combined with Drosophila genetics. This work will be of broad interest to cell biologists interested in autophagy and related cell biology fields.

---

## [Referee Report · Reviewer #1 (Public review)]

Summary:

It is well known that autophagosomes/autolysosomes move along microtubules. However, as these previous studies did not distinguish between autophagosomes and autolysosomes, it remains unknown whether autophagosomes begin to move after fusion with lysosomes or even before fusion. In this manuscript, the authors show using fusion-deficient vps16a RNAi cells that both pre-fusion autophagosomes and lysosomes can move along the microtubules towards the minus end. This was confirmed in snap29 RNAi cells. By screening motor proteins and Rabs, the authors found that autophagosomal traffic is primarily regulated by the dynein-dynactin system and can be counter-regulated by kinesins. They also show that Rab7-Epg5 and Rab39-ema interactions are important for autophagosome trafficking.

Strengths:

This study uses reliable Drosophila genetics and high-quality fluorescence microscopy. The data are properly quantified and statistically analyzed. It is a reasonable hypothesis that gathering pre-fusion autophagosomes and lysosomes in close proximity improves fusion efficiency.

Weaknesses:

(1) This study investigates the behavior of pre-fusion autophagosomes and lysosomes using fusion-incompetent cells (e.g., vps16a RNAi cells). However, the claim that these cells are truly fusion-incompetent relies on citations from previous studies. Since this is a foundational premise of the research, it should be rigorously evaluated before interpreting the data. It's particularly awkward that the crucial data for vps16a RNAi is only presented at the very end of Figure 10-S1; this should be among the first data shown (the same for SNAP29). It would be important to determine the extent to which autophagosomes and lysosomes are fusing (or tethered in close proximity), within each of these cell lines.

(2) In the new Figures 8 and 9, the authors analyze autolysosomes without knocking down Vps16A (i.e., without inhibiting fusion). However, as this reviewer pointed out in the previous round, it is highly likely that both autophagosomes and autolysosomes are present in these cells. This is particularly relevant given that the knockdown of dynein-dynactin, Rab7, and Epg5 only partially inhibits the fusion of autophagosomes and lysosomes (Figure 10H). If the goal is to investigate the effects of fusion, it would be more appropriate to analyze autolysosomes and autophagosomes separately. The authors mention that they can differentiate these two structures based on the size of mCherry-Atg8a structures. If this is the case, they should perform separate analyses for both autophagosomes and autolysosomes.

(3) This is also a continued Issue from the previous review. The authors suggest that autophagosome movement is crucial for fusion, based on the observed decrease in fusion rates in Rab7 and Epg5 knockdown cells (Fig. 10). However, this conclusion is not well supported. It is known that Rab7 and Epg5 are directly involved in the fusion process itself. Therefore, the possibility that the observed decrease is simply due to a direct defect in fusion, rather than an impairment of movement, has not been ruled out.

(4) The term "autolysosome maturation" appears multiple times, yet its meaning remains unclear. Does it refer to autolysosome formation (autophagosome-lysosome fusion), or does it imply a further maturation process occurring after autolysosome formation? This is not a commonly used term in the field, so it requires a clear definition.

(5) In Figure 1-S1D, the authors state that the disappearance of the mCherry-Atg8a signal after atg8a RNAi indicates that the observed structures are not non-autophagic vacuoles. This reasoning is inappropriate. Naturally, knocking down Atg8 will abolish its signal, regardless of the nature of the vacuoles. This does not definitively distinguish autophagic from non-autophagic structures.

---

## [Referee Report · Reviewer #2 (Public review)]

Summary:

This manuscript by Boda et al. describes the results of a targeted RNAi screen in the background of Vps16A-depleted Drosophila larval fat body cells. In this background, lysosomal fusion is inhibited, allowing the authors to analyze the motility and localization specifically of autophagosomes, prior to their fusion with lysosomes to become autolysosomes. In this Vps16A-deleted background, mCherry-Atg8a labeled autophagosomes accumulate in the perinuclear area, through an unknown mechanism.

The authors found that depletion of multiple subunits of the dynein/dynactin complex caused an alternation of this mCherry-Atg8a localization, moving from the perinuclear region to the cell periphery. Interactions with kinesin overexpression suggest these motor proteins may compete for autophagosome binding and transport. The authors extended these findings by examining potential upstream regulators including Rab proteins and selected effectors, and they also examined effects on lysosomal movement and autolysosome size. Altogether, the results are consistent with a model in which specific Rab/effector complexes direct movement of lysosomes and autophagosomes toward the MTOC, promoting their fusion and subsequent dispersal throughout the cell.

Strengths:

Although previous studies of the movement of autophagic vesicles have identified roles for microtubule-based transport, this study moves the field forward by distinguishing between effects on pre- and post-fusion autophagosomes, and by its characterization of the roles of specific Dynein, Dynactin, and Rab complexes in regulating movement of distinct vesicle types. Overall, the experiments are well controlled, appropriately analyzed, and largely support the authors' conclusions..

Weaknesses:

One limitation of the study is the genetic background that serves as basis for the screen. In addition to preventing autophagosome-lysosome fusion, disruption of Vps16A has been shown to inhibit endosomal maturation and to block trafficking of components to the lysosome from both the endosome and Golgi apparatus. Additional effects previously reported by the authors include increased autophagosome production and reduced mTOR signaling. Thus Vps16A-depleted cells have a number of endosome, lysosome and autophagosome-related defects, with unknown downstream consequences. Additionally, the cause and significance of the perinuclear localization of autophagosomes in this background is unclear. Thus, interpretations of the observed reversal of this phenotype are difficult, and have the caveat that they may apply only to this condition, rather than to normal autophagosomes. Additional experiments to observe autophagosome movement or positioning in a more normal environment would improve the manuscript.

Comments on revision:

The revised manuscript and author responses have satisfactorily met my concerns. I have no further issues and congratulate the authors on this work.

---

## [Referee Report · Reviewer #3 (Public review)]

Summary:

In multicellular organisms, autophagosomes are formed throughout the cytosol, while late endosomes/lysosomes are relatively enriched in the perinuclear region. It is known that autophagosomes gain access to the lysosome-enriched region by microtubule-based trafficking. The mechanism by which autophagosomes move along microtubules remains incompletely understood. In this manuscript, Péter Lőrincz and colleagues investigated the mechanism driving the movement of nascent autophagosomes along microtubule towards non-centrosomal microtubule organizing center (ncMTOC) using fly fat body as a model system. The authors took an approach by examining autophagosome positioning in cells where autophagosome-lysosome fusion was inhibited by knocking down the HOPS subunit Vps16A. Despite being generated at random positions in the cytosol, autophagosomes accumulate around the nucleus when Vps16A is depleted. They then performed an RNA interference screen to identify the factors involved in autophagosome positioning. They found that the dynein-dynactin complex is required for trafficking of autophagosomes toward ncMTOC. Dynein loss leads to the peripheral relocation of autophagosomes. They further revealed that a pair of small GTPases and their effectors, Rab7-Epg5 and Rab39-ema, are required for bidirectional autophagosome transport. Knockdown of these factors in Vps16a RNAi cells causes scattering of autophagosomes throughout the cytosol.

Strengths:

The data presented in this study help us to understand the mechanism underlying the trafficking and positioning of autophagosomes.

Weaknesses:

(1) The experiments were performed in Vps16A RNAi KD cells. Vps16A knockdown blocks fusion of vesicles derived from the endolysosomal compartments such as fusion between lysosomes. The pleiotropic effect of Vps16A RNAi may complicate the interpretation.

(2) In this study, the transport of autophagosomes is investigated in fly fat cells. In fat cells, a large number of large lipid droplets accumulate and the endomembrane systems are distinct from that in other cell types. The knowledge gain from this study may not apply to other cell types.

---

## [Author Response]

The following is the authors’ response to the original reviews.

**Reviewer #1 (Public review):**
(1) To distinguish autophagosomes from autolysosomes, the authors used vps16 RNAi cells, which are supposed to be fusion deficient. However, the extent to which fusion is actually inhibited by knockdown of Vps16A is not shown. The co-localization rate of Atg8 and Lamp1 should be shown (as in Figure 8). Then, after identifying pre-fusion autophagosomes and lysosomes, the localization of each should be analyzed.

Thank you for this insightful comment. We analyzed the colocalization of 3xmCherry-Atg8a and GFP-Lamp1, which label autophagic structures and lysosomes, respectively, in Vps16A RNAi fat body cells. As expected, Vps16A silencing markedly reduced the overlap between these two signals, indicating a strong block in autophagosome–lysosome fusion. Moreover, both 3xmCherry-Atg8a and GFP-Lamp1 became more perinuclearly localized compared to the control (luciferase RNAi) cells.

It is also possible that autophagosomes and lysosomes are tethered by factors other than HOPS (even if they are not fused). If this is the case, autophagosomal trafficking would be affected by the movement of lysosomes.

Thank you for raising this possibility. While we cannot fully exclude that autophagosomes might be indirectly transported via tethering to lysosomes, we consider this unlikely. We believe that in Drosophila fat cells, autophagosomes and lysosomes rapidly fuse once in close proximity. Therefore, even if alternative tethering mechanisms exist, they are unlikely to permit prolonged joint trafficking without fusion.

(2) The authors analyze autolysosomes in Figures 6 and 7. This is based on the assumption that autophagosome-lysosome fusion takes place in cells without vps16A RNAi. However, even in the presence of Vps16A, both pre-fusion autophagosomes and autolysosomes should exist. This is also true in Figure 8H, where the fusion of autophagosomes and lysosomes is partially suppressed in knockdown cells of dynein, dynactin, Rab7, and Epg5. If the effect of fusion is to be examined, it is reasonable to distinguish between autophagosomes and autolysosomes and analyze only autolysosomes.

Thank you for this careful observation. The 3xmCherry-Atg8a reporter is well suited to identify both autophagosomes and autolysosomes, as the mCherry fluorophore is resistant to degradation in the acidic environment of autolysosomes. Notably, mCherry-Atg8a–positive autolysosomes appear larger and brighter than pre-fusion autophagosomes, which are typically smaller and dimmer, especially under fusion-deficient conditions (e.g., Figure 4). Therefore, we use these morphological differences as a proxy to distinguish between the two.

To improve structural assignment, we incorporated endogenous Lamp1 staining (Figure 10) and a Lamp1-GFP reporter (Figure 10—figure supplement 1). Vesicles positive for mCherryAtg8a but negative for Lamp1 are considered pre-fusion autophagosomes. Structures double-positive for mCherry-Atg8a and Lamp1 represent autolysosomes, while Lamp1positive, Atg8a-negative vesicles correspond to non-autophagic lysosomes. To clarify these interpretations, we revised the Results section and explained these reporters in more detail.

(3) In this study, only vps16a RNAi cells were used to inhibit autophagosome-lysosome fusion. However, since HOPS has many roles besides autophagosome-lysosome fusion, it would be better to confirm the conclusion by knockdown of other factors (e.g., Stx17 RNAi).

Thank you for this valuable suggestion. We initially considered using Syntaxin17 RNAi; however, our recent findings indicate that loss of Syx17 results in a HOPS-dependent tethering lock between autophagosomes and lysosomes (DOI: 10.1126/sciadv.adu9605). In this case, tethered vesicles would likely move together, confounding the interpretation of autophagosome-specific trafficking.

Therefore, we turned to other SNAREs such as Vamp7 and Snap29. One Snap29 RNAi was located on the appropriate chromosome needed for our genetic experiments. We generated a transgenic fly line expressing both Snap29 RNAi and the mCherry-Atg8a reporter under a fat body-specific R4 promoter. When we tested our key trafficking hits in this background, we observed similar autophagosome localization phenotypes as in Vps16A RNAi cells. These results, now included in the revised manuscript (see Figure 6), confirm that the observed transport phenotypes are not specific to Vps16A or HOPS complex loss.

(4) Figure 8: Rab7 and Epg5 are also known to be directly involved in autophagosomelysosome tethering/fusion. Even if the fusion rate is reduced in the absence of Rab7 and Epg5, it may not be the result of defective autophagosome movement, but may simply indicate that these molecules are required for fusion itself. How do the authors distinguish between the two possibilities?

Thank you for this important point. While Rab7 and Epg5 indeed participate in autophagosome–lysosome tethering and fusion, our data suggest they also contribute to autophagosome movement. This is evident from the distinct phenotypes observed upon Rab7 or Epg5 RNAi compared to Vps16A or SNARE RNAi. Depletion of Vps16A, Syx17, Vamp7, or Snap29 (factors involved specifically in fusion) results in perinuclear accumulation of autophagosomes. In contrast, Rab7 or Epg5 RNAi leads to a dispersed autophagosome pattern throughout the cytoplasm.

These differences suggest that Rab7 and Epg5 play additional roles in positioning autophagosomes. Supporting this, our co-immunoprecipitation experiments show that Epg5 interacts with dynein motors. Therefore, we propose that Rab7 and Epg5 influence both autophagosome fusion and their microtubule-based transport.

**Reviewer #2 (Public review):**
One limitation of the study is the genetic background that serves as the basis for the screening. In addition to preventing autophagosome-lysosome fusion, disruption of Vps16A has been shown to inhibit endosomal maturation and block the trafficking of components to the lysosome from both the endosome and Golgi apparatus. Additional effects previously reported by the authors include increased autophagosome production and reduced mTOR signaling. Thus Vps16A-depleted cells have a number of endosome, lysosome, and autophagosome-related defects, with unknown downstream consequences. Additionally, the cause and significance of the perinuclear localization of autophagosomes in this background is unclear. Thus, interpretations of the observed reversal of this phenotype are difficult, and have the caveat that they may apply only to this condition, rather than to normal autophagosomes. Additional experiments to observe autophagosome movement or positioning in a more normal environment would improve the manuscript.

Thank you for highlighting this limitation. We have tried to conduct time-lapse imaging of live fat body cells expressing 3xmCherry-Atg8a and GFP-Lamp1 to visualize the movement and fusion events of pre-fusion autophagosomes (3xmCherry-Atg8a positive and GFP-Lamp1 negative) and lysosomes (GFP-Lamp1 positive). Despite different experimental setups and durations of starvation, no vesicle movement was observed at all, so live imaging of larval Drosophila fat tissue will require time-consuming optimizations of in vitro culture conditions. Consistent with this, we did not find any literature data where organelle motility in fat body cells was successfully observed. Nuclear positioning in fat body cells was investigated in detail in an excellent study, however the authors were able to observe only very little movement of the nuclei by live imaging (Zheng et al. Nat Cell Biol. 2020 Mar;22(3):297-309. doi: 10.1038/s41556-020-0470-7), further highlighting the technical difficulties of live or timelapse imaging in this tissue type.

Specific comments(1) Several genes have been described that when depleted lead to perinuclear accumulation of Atg8-labeled vesicles. There seems to be a correlation of this phenotype with genes required for autophagosome-lysosome fusion; however, some genes required for lysosomal fusion such as Rab2 and Arl8 apparently did not affect autophagosome positioning as reported here. Thus, it is unclear whether the perinuclear positioning of autophagosomes is truly a general response to disruption of autophagosome-lysosome fusion, or may reflect additional aspects of Vps16A/HOPS function. A few things here would help. One would be an analysis of Atg8a vesicle localization in response to the depletion of a larger set of fusionrelated genes. Another would be to repeat some of the key findings of this study (effects of specific dynein, dynactin, rabs, effectors) on Atg8a localization when Syx17 is depleted, rather than Vps16A. This should generate a more autophagosome-specific fusion defect.

Thank you for this insightful suggestion. We recently discovered that Syx17 depletion induces a HOPS-dependent tethering lock between autophagosomes and lysosomes (DOI: 10.1126/sciadv.adu9605), making it unsuitable for modeling autophagosome-specific fusion defects. In contrast, Vamp7 and Snap29 knockdowns do not appear to cause such tethering lock. We were able to generate a suitable Drosophila line using a Snap29 RNAi transgene located on a compatible chromosome. Upon testing key hits from our screen in this background, we found that autophagosomes redistributed similarly, supporting our conclusions. These new results have been included in the revised manuscript (see Figure 6)

Third, it would greatly strengthen the findings to monitor pre-fusion autophagosome localization without disrupting fusion. Such vesicles could be identified as Atg8a-positive Lamp-negative structures. The effects of dynein and rab depletion on the tracking of these structures in a post-induction time course would serve as an important validation of the authors' findings.

Thank you for this helpful suggestion. As described above, we attempted time-lapse imaging of 3xmCherry-Atg8a and GFP-Lamp1-expressing fat body cells under various conditions to identify motile pre-fusion autophagosomes. However, we did not observe any vesicle movement, regardless of the starvation duration or experimental setup. As this likely reflects technical limitations of ex vivo fat body imaging, we were unable to achieve live tracking of autophagosome dynamics without introducing perturbations. This limitation is now discussed in the revised manuscript.

(2) The authors nicely show that depletion of Shot leads to relocalization of Atg8a to ectopic foci in Vps16A-depleted cells; they should confirm that this is a mislocalized ncMTOC by colabeling Atg8a with an MTOC component such as MSP300. The effect of Shot depletion on Atg8a localization should also be analyzed in the absence of Vps16A depletion.

Thank you for this positive comment. We co-labeled Atg8a with the minus-end microtubule marker Khc-nod-LacZ in both shot single knockdown and shot; vps16A double knockdown cells. Ectopic Khc-nod-LacZ-positive MTOC foci were clearly visible in both conditions, and Atg8a-positive autophagosomes accumulated around these structures. These findings confirm that Shot depletion induces ectopic MTOC formation, which correlates with autophagosome relocalization. The new data have been incorporated into the revised manuscript (see Figure 1O-S).

(3) The authors report that depletion of Dynein subunits, either alone (Figure 6) or codepleted with Vps16A (Figure 2), leads to redistribution of mCherry-Atg8a punctae to the "cell periphery". However, only cell clones that contact an edge of the fat body tissue are shown in these figures. Furthermore, in these cells, mCherry-Atg8a punctae appear to localize only to contact-free regions of these cells, and not to internal regions of clones that share a border with adjacent cells. Thus, these vesicles would seem to be redistributed to the periphery of the fat body itself, not to the periphery of individual cells. Microtubules emanating from the perinuclear ncMTOC have been described as having a radial organization, and thus it is unclear that this redistribution of mCherry-Atg8a punctae to the fat body edge would reflect a kinesin-dependent process as suggested by the authors.

Thank you for this detailed observation. We frequently observe autophagosomes accumulating in contact-free peripheral regions of dynein-depleted cells, resulting in an asymmetric distribution. While previous studies describe a radial microtubule organization in fat body cells, none of them directly label MT plus ends, the direction of kinesin-based transport.

To further explore this, we overexpressed a HA-tagged kinesin, Klp98A-3xHA, in both control and Vps16A RNAi backgrounds. Immunolabeling revealed that Klp98A localizes to the contact-free peripheral regions in both conditions, consistent with the distribution of autophagosomes in dynein knockdown cells. This supports our interpretation that kinesindependent transport drives autophagosome redistribution in the absence of dynein, and that fat body cells exhibit subtle asymmetries in MT polarity that influence this transport. These new results have been included in the revised manuscript (see Figure 3G, H).

(4) To validate whether the mCherry-Atg8a structures in Vps16A-depleted cells were of autophagic origin, the authors depleted Atg8a and observed a loss of mCherry- Atg8a signal from the mosaic cells (Figure S1D, J). A more rigorous experiment would be to deplete other Atg genes (not Atg8a) and examine whether these structures persist.

Thank you for the suggestion to further validate our reporter. We depleted Atg1, a key kinase required for phagophore initiation, in the Vps16A RNAi background. This completely abolished the punctate mCherry-Atg8a distribution in knockdown cells (see Figure 1—figure supplement 1E, K), confirming that the labeled structures are indeed of autophagic origin.

(5) The authors found that only a subset of dynein, dynactin, rab, and rab effector depletions affected mCherry-Atg8a localization, leading to their suggestion that the most important factors involved in autophagosome motility have been identified here. However, this conclusion has the caveat that depletion efficiency was not examined in this study, and thus any conclusions about negative results should be more conservative.

Thank you for this constructive feedback. We agree that negative results must be interpreted conservatively due to potential differences in knockdown efficiency. We have revised our conclusions accordingly, clarifying that the factors identified are key for autophagosome motility, while acknowledging the possibility of false negatives.

**Reviewer #3 (Public review):**
Major concerns:(1) The localization of EPG5 should be determined. The authors showed that EPG5 colocalizes with endogenous Rab7. Rab7 labels late endosomes and lysosomes. Previous studies in mammalian cells have shown that EPG5 is targeted to late endosomes/lysosomes by interacting with Rab7. EPG5 promotes the fusion of autophagosomes with late endosomes/lysosomes by directly recognizing LC3 on autophagosomes and also by facilitating the assembly of the SNARE complex for fusion. In Figure 5I, the EPG5/Rab7colocalized vesicles are large and they are likely to be lysosomes/autolysosomes.

Thank you for suggesting to improve our Epg5 localization data. We performed triple immunostaining for Atg8a, Lamp1-3xmCherry, and Epg5-9xHA in S2R+ cells. In addition to triple-positive structures—likely representing autolysosomes—we observed Atg8a and Epg59xHA double-positive vesicles that lacked Lamp1-3xmCherry signal, which likely correspond to pre-fusion autophagosomes. Based on these results, we propose that in addition to arriving via the endocytic route, Epg5 may also reach lysosomes through autophagosomes. These findings have been included in the revised manuscript (see Figure 5K).

(2) The experiments were performed in Vps16A RNAi KD cells. Vps16A knockdown blocks fusion of vesicles derived from the endolysosomal compartments such as fusion between lysosomes. The pleiotropic effect of Vps16A RNAi may complicate the interpretation. The authors need to verify their findings in Stx17 KO cells, as it has a relatively specific effect on the fusion of autophagosomes with late endosomes/lysosomes.

Thank you for this valuable suggestion. We initially considered Syntaxin17 for validation; however, we recently found that loss of Syx17 leads to a HOPS-dependent tethering lock between autophagosomes and lysosomes, which would confound interpretation, as autophagosomes remain tethered to lysosomes (DOI: 10.1126/sciadv.adu9605). Therefore, Syntaxin17 loss is not suitable for our purpose. Among the remaining fusion SNAREs, one RNAi line targeting Snap29 was available on a compatible chromosome for generating Drosophila lines equivalent to those used in the screen. We established this Snap29 RNAicontaining tester line and crossed it with our top hits. We observed that autophagosome motility was comparable to that in the Vps16A RNAi background, further supporting our conclusions. These results have been incorporated into the revised manuscript (see Figure 6)

(3) Quantification should be performed in many places such as in Figure S4D for the number of FYVE-GFP labeled endosomes and in Figures S4H and S4I for the number and size of lysosomes.

Thank you for pointing this out. We performed the suggested quantifications and statistical analyses for FYVE-GFP labeled endosomes, as well as for the number and size of lysosomes. The updated data are now presented in the revised Figure 5—figure supplement 1.

(4) In this study, the transport of autophagosomes is investigated in fly fat cells. In fat cells, a large number of large lipid droplets accumulate and the endomembrane systems are distinct from that in other cell types. The knowledge gained from this study may not apply to other cell types. This needs to be discussed.

Thank you for raising this important point. We agree that our findings may not be fully generalizable to all cell types. Given that the organization of the microtubule network depends on both cell function and developmental stage, it is plausible that the molecular machinery described here operates differently elsewhere. We now mention this limitation in the Discussion.

Minor concerns:(5) Data in some panels are of low quality. For example, the mCherry-Atg8a signal in Figure 5C is hard to see; the input bands of Dhc64c in Figure 5L are smeared.

Thank you for pointing this out. We repeated the experiment shown in Figure 5C and replaced the panel with a clearer image. The smeared Dhc64C input bands in Figure 5L result from the unusually large size of this protein, which affects its electrophoretic migration. We mentioned this point in the corresponding figure legend.

(6) In this study, both 3xmCherry-Atg8a and mCherry-Atg8a were used. Different reporters make it difficult to compare the results presented in different figures.

Thank you for this comment. Both 3xmCherry-Atg8a and mCherry-Atg8a are well-established reporters that behave similarly as autophagic markers. Nevertheless, to avoid confusion, we ensured that each figure uses only one type of reporter consistently, which is now clearly indicated in the revised manuscript.

(7) The small autophagosomes presented in Figures such as in Figure 1D and 1E are not clear. Enlarged images should be presented.

Thank you for your suggestion. We repeated these experiments and replaced the relevant panels with higher-quality images, including enlarged insets to better visualize small autophagosomes. These updated figures are now included in the revised manuscript.

(8) The authors showed that Epg5-9xHA coprecipitates with the endogenous dynein motor Dhc64C. Is Rab7 required for the interaction?

Thank you for this insightful question. We tested this by co-transfecting S2R+ cells with Epg5-9xHA and different forms of Rab7: wild-type, GTP-locked (constitutively active), and GDP-locked (dominant-negative). Our results indicate that the strength of Epg5-Dhc interaction does not change in the presence of either GTP-locked or GDP-locked Rab7. However, we believe that Epg5 and dynein are recruited to the vesicle membranes via Rab7 in vivo, so we did not include these results in the revised manuscript.

(9) The perinuclear lysosome localization in Epg5 KD cells has no indication that Epg5 is an autophagosome-specific adaptor.

Thank you for this important comment. Accordingly, we have toned down our statements about Epg5 functions throughout the revised manuscript.

**Reviewer #1 (Recommendations for the authors):**
(1) Figure 6: What do "autolysosome maturation" and "small autolysosomes" mean? Do different numbers of lysosomes fuse to a single autophagosome?

Thank you for highlighting this point. We concluded that the formation of smaller autolysosomes—compared to controls—is likely due to a defect in autolysosome maturation, as is often the case. We had not explicitly considered whether a different number of lysosomes fuse with each autophagosome during this process. We clarified this issue in the revised manuscript.

(2) Figure 5A shows that the localization of endogenous Atg8 requires Epg5, but the data is not as clear as for mCherry-Atg8 (Figure 4B). Why the difference?

Thank you for this question. The difference arises because the mCherry-Atg8a reporter strongly labels autolysosomes, as the mCherry fluorophore remains stable in acidic compartments. As a result, mCherry-Atg8a labels both autophagosomes and autolysosomes, but the strong autolysosomal signal originating from the surrounding GFP negative, nonRNAi cells can make accumulated autophagosomes appear fainter in fusion-defective cells (as in Figure 4). In contrast, endogenous Atg8a is degraded in lysosomes, and therefore labels only autophagosomes. This means that the appearance of these two experiments can be slightly different, but since in both cases autophagosomes no longer accumulate in the perinuclear region of Vps16A,Epg5 double RNAi cells we can conclude that Epg5 is required for autophagosome positioning. We explained this difference of the two methods in the revised manuscript where it first appears (Figure 1B and Figure 1—figure supplement 1A).

(3) Blue letters on the black micrographs are hard to see. Some of the other letters are also small and hard to read.

Thank you for this suggestion. We improved the visibility and readability of the labels in the revised figures.